# Southern Ocean evidence for recurring West Antarctic Ice Sheet destabilization during Marine Isotope Stage 11

L. Jebasinski [1] ✉, D. A. Frick [1], A. K. I. U. Kapuge[2], C. Basak[2], M. Saavedra-Pellitero [3], G. Winckler [4], F. Lamy [5] & J. Gottschalk [1]

Millennial-scale deoxygenation of Antarctic Bottom Water (AABW) in the Atlantic Southern Ocean during past interglacials was linked to West Antarctic Ice Sheet (WAIS) melt-driven suppression of dense water formation along the Antarctic margin. However, the circum-Antarctic extent of these 'AABW stagnation events' and drivers of WAIS retreat remain unclear. Here, we identify recurring bottom water $O_2$ minima in the central Pacific Southern Ocean during Marine Isotope Stage (MIS) 11 (424–374 ka ago) that are synchronous with their Atlantic Southern Ocean counterparts. As they (partially) align with Circumpolar Deep Water (CDW) warming above present-day levels and/or a reorganization of deep-ocean circulation, we postulate recurring and synchronized Pacific-Atlantic AABW perturbation events during MIS11 through WAIS retreat and enhanced exposure to ocean heat (i.e., CDW) from below. This indicates a significant contribution of WAIS meltwater to sea-level highstands during MIS11 and, by analogy, to sea-level rise due to ocean warming in the future.

Observations show that the Antarctic Ice Sheet (AIS) has lost mass over the past decades[1,2]. The West Antarctic Ice Sheet (WAIS) is particularly affected as it is prone to instabilities due to below sea-level grounding[3], a landward-sloping geometry of the bedrock[4], over-deepened continental shelf areas[5], and large ice shelf areas rimming the ice sheet[2,6]. When fully melted, WAIS has the potential to raise global sea-levels by up to 4.3 m eustatic sea-level equivalent[7]. Antarctic Bottom Water (AABW) that forms by sinking of dense oxygenated shelf waters along the Antarctic margin supplies oxygen to the ocean interior[8], and is therefore intricately linked to the geometry and/or advance of the WAIS on the Antarctic continental shelf[9]. Instrumental records bear witness to hydrographic and geochemical changes in the Southern Ocean that manifest the impact of WAIS retreat and associated meltwater supply on the Southern Ocean[9]: Antarctic shelf waters became fresher[10], and AABW contracted[11], lost oxygen[12] and became less dense due to combined effects of warming and freshening[13]. Continued

Antarctic ice mass loss, and hence meltwater supply to the Southern Ocean, is expected for the coming decades and centuries[14], which may contribute to global sea-level rise[15] and alter ocean circulation dynamics[16-18].

One of the primary causes of WAIS destabilization from below via ocean warming is the upwelling of warm Circumpolar Deep Water (CDW) underneath vast floating ice shelves[19,20] such as the Ross Ice Shelf and/or the Filchner-Ronne Ice Shelf (Fig. 1)[6,15]. Antarctic ice shelves act as a stabilization band, buttressing hinterland ice sheet flow across the grounding line[2]. However, they also represent a band of vulnerability making AIS prone to melting from below[6]. Upwelling of warm CDW near the Antarctic periphery favors WAIS melt and leads to meltwater discharge into the Southern Ocean in the Ross Sea, the Weddell Sea and the Amundsen-Bellingshausen Sea as highlighted by observations[10,21] and numerical simulations[22]. Several processes can control the heat supply from the Southern Ocean to the WAIS margin,

[1]Institute of Geosciences, Kiel University, Kiel, Germany. [2]Department of Earth Sciences, University of Delaware, Newark, DE, USA. [3]School of the Environment and Life Sciences, University of Portsmouth, Portsmouth, UK. [4]Columbia Climate School, Columbia University, New York, NY, USA. [5]Alfred Wegener Institute (AWI) Helmholtz Centre for Polar and Marine Research, Bremerhaven, Germany. ✉e-mail: lena.jebasinski@ifg.uni-kiel.de

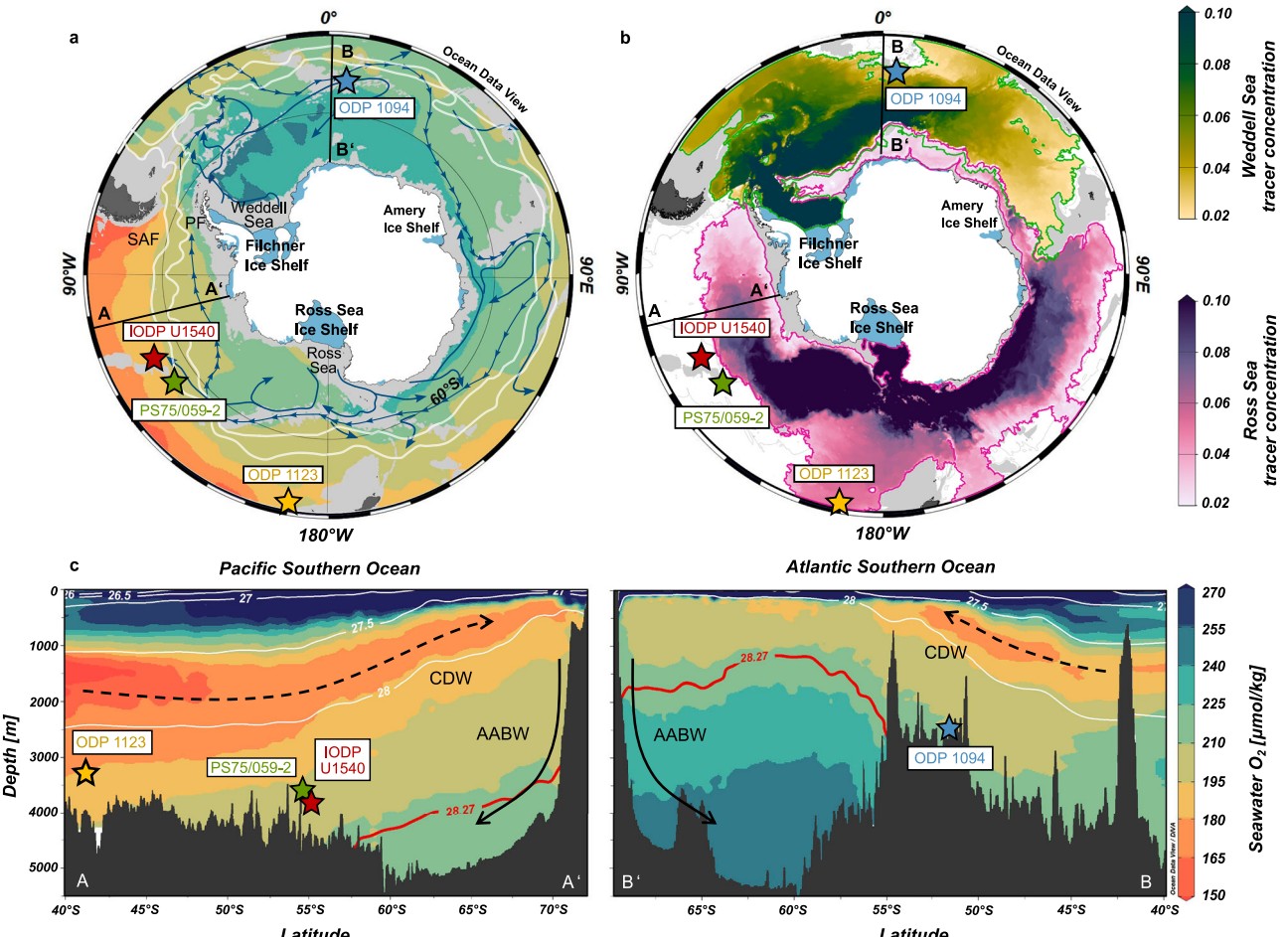

**Fig. 1 | Southern Ocean circulation dynamics. a** Seawater $O_2$ concentrations (in μmol/kg) at 3 km water depth according to the Global Ocean Data Analysis Project version 2 (GLODAPv2)[48]. White lines (from south to north) indicate the Polar Front (PF) and the South Antarctic Front (SAF)[83]. Blue arrows show the flow of Antarctic Bottom Water (AABW) from its source regions[8]. **b** Simulated present-day AABW tracer concentrations at the model bottom cell (after 61 model years) emerging from the Weddell Sea (upper color bar) and the Ross Sea (lower color bar) based on the ACCESS-OM2-01 model[25] with green and pink contours showing areas where tracer concentrations are larger than 0.02, respectively, highlighting the dispersal of AABW varieties in the Southern Ocean controlled by bathymetry. Light gray regions in **a**, **b** indicate bathymetry shallower than 3 km water depth. Blue areas in Antarctica highlight major ice shelves. **c** Water column transects in the Pacific (AA' in **a**, **b**) and Atlantic Southern Ocean (BB' in **a**, **b**) of seawater $O_2$ concentrations (shading) and neutral densities (white and red contours) (GLODAPv2)[48]. The neutral density $\gamma_n = 28.27$ kg/m³ separating Circumpolar Deep Water (CDW) from AABW is shown in red[8]. Arrows indicate the flow paths of CDW (stippled) and AABW (solid). The locations of IODP Site U1540, ODP Site 1094, ODP Site 1123 and core PS75/059-2 are indicated with stars. Figure was created using Ocean Data View[84].

such as changes in the formation of Dense Shelf Water (DSW) and the associated density gradients at the continental shelf break keeping warm CDW off Antarctic continental shelves[20], adjustments in wind- and gyre circulation propelling CDW across the Antarctic shelf break[23], variations in the heat content and/or upwelling rate of CDW offshore[24], and/or combinations thereof. Observations[10] and numerical models[12,16,18] attest that freshwater fluxes from WAIS melt can significantly perturb AABW formation along the Antarctic margin, particularly in the Ross- and Weddell Sea, where the majority of deep water mass formation in the Southern Ocean occurs today[8,25]. Yet, observational evidence on the interplay between WAIS- and CDW dynamics for different climate boundary conditions in the past remains scarce.

Marine sediment core evidence from Ocean Drilling Program (ODP) Site 1094 in the deep Atlantic Southern Ocean reveals transient decreases in bottom water $O_2$ (BWO) levels during late Pleistocene interglacials[26,27]. Although some of these phases of low BWO conditions were explained by increases in surface ocean productivity and concomitant respiratory BWO consumption[16], organic carbon fluxes can be excluded as the main driver of observed BWO minimum events during Marine Isotope Stage (MIS) 5e at ~127 kilo years before present (ka BP)[27] and during MIS11 at ~397 ka BP (Fig. 2)[26]. Instead, these transient lows in BWO levels were found to be linked to short-term disruptions (i.e., a slowdown) in AABW formation in the Weddell Sea Embayment and an associated decline in the dispersal of well-oxygenated AABW northward[26,27]. These events were thus dubbed 'AABW stagnation events'[27] and were suggested to be driven by melt-water supply to the Weddell Sea[26,27]. However, transient AABW deoxygenation events during warmer-than-interglacial conditions have so far not been identified outside of the Atlantic Southern Ocean, specifically at a sediment core location other than ODP Site 1094. While WAIS retreat and associated freshwater input into the Southern Ocean were linked to the observed 'AABW stagnation events' in the Atlantic Southern Ocean, the spatial extent, and drivers of these events as well as possible associated ocean temperature thresholds of WAIS melt remain largely unconstrained.

Here, we reconstruct BWO concentrations in the central South Pacific Ocean during MIS11 based on high-resolution reconstructions of authigenic U (aU) enrichments in foraminiferal coatings in sediment cores from International Ocean Discovery Program (IODP) Site U1540 (55°08.467'S, 114°50.515'W; 3580 m water depth; Fig. 1). Our study site is presently bathed in Lower CDW, which forms through the entrainment of well-oxygenated Ross Sea-derived AABW

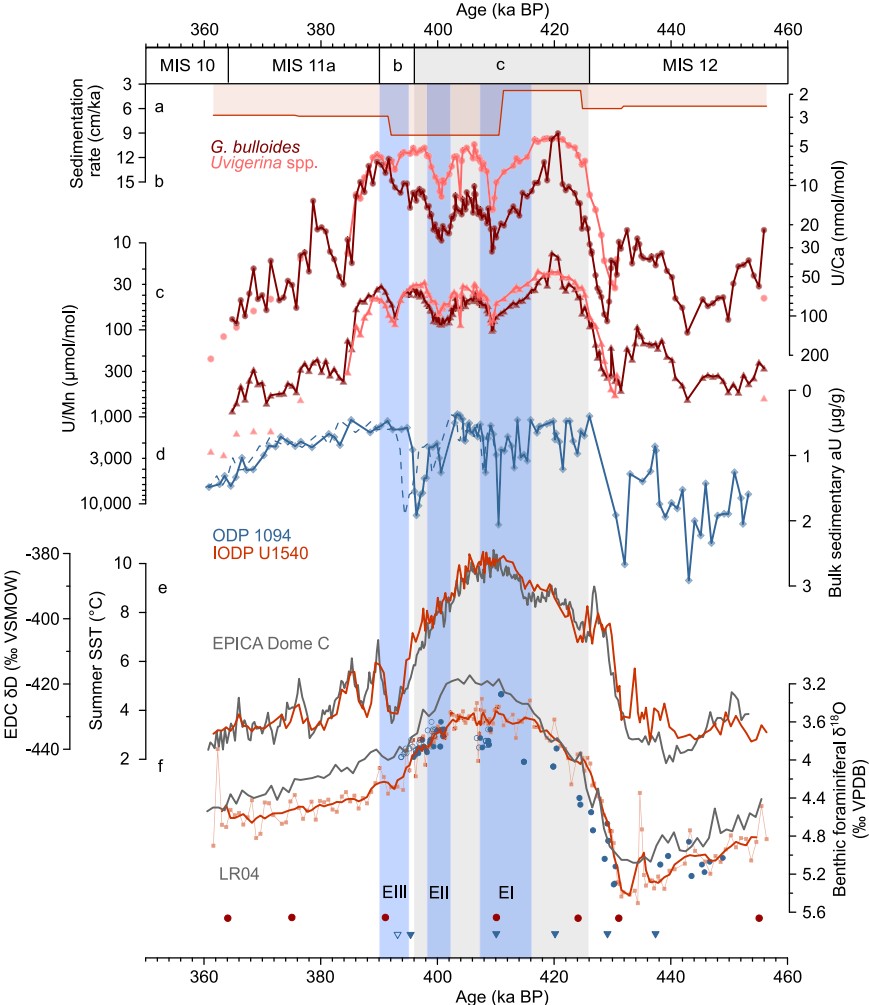

**Fig. 2 | Bottom water oxygenation changes in the central Pacific- and Southern Ocean during Marine Isotope Stage (MIS) 12 to 10. a** Sedimentation rate (cm/ka) at IODP Site U1540. *Globigerina bulloides* (dark red) and *Uvigerina* spp. (light red) **b** U/Ca and **c** U/Mn ratios at IODP Site U1540. d) Bulk sedimentary authigenic U (aU) levels[26] at ODP Site 1094. The stippled line indicates the aU record according to an adjusted age model (see Results). **e** Summer sea surface temperature (SST) estimates based on *N. pachyderma* abundances at IODP Site U1540 (red)[73] and water isotope (δD) variations of the Antarctic EPICA Dome C (EDC) ice core[30] (gray, on the AICC2023 ice age scale[38]). **f** Stratigraphic alignment of

*C. kullenbergi* and *C. wuellerstorfi* δ[18]O variations at IODP Site U1540 (red) and *Cibicidoides* spp. δ[18]O variations at ODP Site 1094 (blue)[36] to the global benthic foraminiferal LR04 δ[18]O stack (gray)[37] with indication of tiepoints between them at the bottom (circles: IODP Site U1540; triangles[36]: ODP Site 1094). Open symbol shows the adjusted tiepoint for ODP Site 1094 (see Results). MIS and substages are indicated following ref. 85, with MIS11c indicated by the gray box[28]. Transient peaks in foraminiferal **b** U/Ca and **c** U/Mn levels at IODP Site U1540 during MIS11 are highlighted with blue boxes labeled with roman numbers, i.e., deoxygenation events (E) EI-EIII.

(Fig. 1a, b)[8,25]. This makes IODP Site U1540 particularly sensitive to changes in the oxygenation and/or formation of AABW in the Pacific sector of the Southern Ocean and their possible mechanistic link to WAIS retreat. We specifically focus on the climatic optimum of MIS11, i.e., MIS11c (~426–396 ka BP)[28], because it is often considered one of the best analogs for the current interglacial due to several reasons. First, both MIS11c and the Holocene are characterized by low eccentricity-driven insolation forcing[28]. Second, the protracted duration of MIS11c (>20 ka)[28] matches the predicted length of the Holocene under unabated anthropogenic greenhouse gas emissions[29], and might therefore allow for an assessment of the WAIS response to an extended exposure to ocean heat. Third, global mean temperatures during MIS11c were up to 2 °C higher than pre-industrial levels[28,30], matching predictions for the end of this century[31]. Finally, sea-level high-stands during MIS11c reached ~6–13 m above present-day levels and suggest significant contribution from the Antarctic ice sheet[32,33]. Our multi-proxy approach allows an assessment of the drivers governing reconstructed BWO changes at IODP Site U1540 based on combined reconstructions of

surface ocean productivity using high-resolution X-ray fluorescence (XRF) Ba/Fe variations and opal (i.e., biogenic silica) weight percentages, deep-water mass provenance changes based on (biology-independent) Nd isotope variations (ε_Nd) of fossil fish teeth/-debris and planktic foraminifera, and bottom water (i.e., CDW) temperature (BWT) estimates via *Uvigerina* spp. Mg/Ca ratios–thereby circumventing stratigraphic biases between proxy records. A comparison of our data with ODP Site 1094 in the Atlantic sector of the Southern Ocean[26], a site presently bathed in Lower CDW primarily influenced by Weddell Sea-derived AABW (Fig. 1a, b)[8,25], provides evidence for recurring and synchronous AABW deoxygenation events in the Pacific and Atlantic sectors of the Southern Ocean during MIS11c and 11b. This suggests widespread WAIS destabilization both in the Weddell- and Ross Sea Embayment linked to the long excess warm climate conditions of MIS11, leading to meltwater-driven interruptions in deep water formation in the Ross Sea and Weddell Sea. Given the synchrony of these AABW deoxygenation and/or -contraction events, we postulate a significant contribution of WAIS retreat to global sea-level high stands during MIS11.

## Results

### Bottom water oxygen changes

The enrichment of aU in foraminiferal coatings indicated by foraminiferal U/Ca and U/Mn ratios have been suggested to reflect redox-driven changes in aU enrichment in marine sediments, and hence BWO levels[34]. *Globigerina bulloides* and *Uvigerina* spp. U/Ca- and U/Mn ratios at IODP Site U1540 during MIS10 and 12 are higher than during MIS11, which is consistent with glacial-interglacial variations in sedimentary aU levels in the Atlantic Southern Ocean, including ODP Site 1094 (Fig. 2b–d)[26,34]. During MIS11, foraminiferal U/Ca and U/Mn ratios are punctuated by recurring, millennial-scale increases in U/Ca by ~20 nmol/mol (*G. bulloides*) and ~10 nmol/mol (*Uvigerina* spp.) as well as rises in U/Mn by ~60 μmol/mol (*G. bulloides*) and ~50 μmol/mol (*Uvigerina* spp.), specifically at ~416–407 ka BP, ~402–398 ka BP and ~395–390 ka BP (albeit with a poor expression of the youngest event in *G. bulloides* U/Ca levels; Fig. 2b, c). The magnitude of these foraminiferal aU enrichments at IODP Site U1540 is roughly a quarter of the MIS12-to-MIS11 transition (Fig. 2b, c). Although changes in seawater carbonate ion ($[CO_3^{2-}]$) variations alter U/Ca ratios in foraminiferal tests during calcification[35], *G. bulloides* U/Ca shifts of ~20 nmol/mol suggest a surface ocean $[CO_3^{2-}]$ change of more than 400 μmol/kg, which is unrealistic. Similar U/Ca-$[CO_3^{2-}]$-calibrations for *Uvigerina* spp. do not exist. However, it is unlikely that both the planktic foraminifer *G. bulloides* and the benthic foraminifer *Uvigerina* spp. record simultaneous changes in surface and bottom water $[CO_3^{2-}]$, respectively, especially with the extreme magnitude as implied by existing calibrations[35]. In contrast, the high U/Ca values recorded at IODP Site U1540 suggest aU enrichment in authigenic foraminiferal coatings to be the main driver of our foraminiferal U/Ca- and U/Mn data. The fact that both planktic and benthic foraminiferal U/Mn records (and U/Ca with potential biases from different test morphometrics[34]) closely agree with each other reinforces our notion of a primary authigenic influence of BWO variations and sedimentary redox-conditions on our aU-based proxies. We thus refer to the millennial-scale increases in U/Ca and U/Mn levels at IODP Site U1540 at ~416–407, ~402–398 and ~395–390 ka BP as "deoxygenation events" (hereafter referred to as EI, EII and EIII, respectively, Fig. 2), in analogy to the definition of "stagnation event" of ref. 27; yet, we deliberately choose to refer to EI-EIII without a mechanistic implication.

The first two increases in foraminiferal U/Ca and U/Mn levels during MIS11c during EI and EII match within age uncertainties with millennial-scale peaks in bulk sedimentary aU levels at ODP Site 1094 in the Southern Atlantic Ocean[26], while the agreement is less clear for EIII (Fig. 2b–d). The age model of ODP Site 1094, which is based on benthic foraminiferal $\delta^{18}O$ tuning[36] to the LR04 benthic $\delta^{18}O$ stack[37] (similar to IODP Site U1540; Methods), is linked with high age uncertainties given sparse benthic $\delta^{18}O$ constraints in the study interval (Fig. 2f). While the chronology at IODP Site U1540 is confirmed by an excellent agreement of *Neogloboquadrina pachyderma* abundance-based summer sea surface temperature (SST) estimates at the study site with water isotope variations ($\delta D$) in the Antarctic EPICA Dome C ice core[30] based on the AICC2023 ice age scale[38] (Fig. 2e), a similar independent test is lacking for ODP Site 1094. An adjustment of the existing age model at ODP Site 1094 within its uncertainties and adhering to the original age model premise[36] emphasizes ambiguities regarding the relative timing of EIII at IODP Site U1540 and ODP Site 1094 (Fig. 2f). Specifically, considering a slight shift of the tiepoint at 395.2 ka BP at ODP Site 1094 (by ~2 ka) maintaining a good match between the benthic foraminiferal $\delta^{18}O$ record to the LR04 $\delta^{18}O$ stack, shifts the aU enrichment event at ODP Site 1094 at 397 ka BP by ~2 ka towards younger ages (Fig. 2d, f). This sensitivity test highlights that the foraminiferal U/Ca and U/Mn increase during MIS11b at IODP Site U1540 (i.e., EIII) might have coincided with a similar aU enrichment event at ODP Site 1094 (Fig. 2d). Without high-resolution $\delta^{18}O$ data and/or additional age

control for ODP Site 1094 it can, however, not be excluded that EIII represents a local event at IODP Site U1540.

### Surface ocean productivity changes

To deconvolve possible biological, dynamical and/or physical influences on BWO variations at the study site, we use for comparability the same marine sediment core proxy as applied at ODP Site 1094, i.e., high-resolution XRF log(Ba/Fe) ratios (Fig. 3d), to approximate the flux of organic matter to the sediment[39], in addition to temporal changes in opal weight percentages[40]. At IODP Site U1540, XRF log(Ba/Fe) ratios are higher during MIS11 than during the adjacent glacial intervals MIS10 and 12, showing similar trends[39] as seen at ODP Site 1094 (Fig. 3d). These trends are consistent with an increase in coccolith accumulation rates and temperature-corrected coccolith Sr/Ca ratios during MIS11 in nearby South Pacific core PS75/059-2 (Fig. 3e), which reflect enhanced coccolithophore productivity in the central South Pacific during that time[41]. However, opal weight percentages at IODP Site U1540 are lower during MIS11 than during MIS12 and MIS10, which is consistent with opal content data from core PS75/059-2 (Fig. 3f)[40,42]. While this may indicate major shifts in ecosystem structures (i.e., siliceous versus calcareous organisms) over glacial-interglacial timescales, systematic, marked increases in surface ocean productivity, and by inference the export of organic matter to the seafloor, are not observed during periods of elevated *G. bulloides* and *Uvigerina* spp. U/Ca and U/Mn ratios at IODP Site U1540 (i.e., EI-EIII; Fig. 3b).

### Variations in deep water provenance

Given the uncertainties in unraveling the drivers governing BWO variations from reconstructions of export production changes alone, we combine our analyses with Nd isotope (i.e., $^{143}Nd/^{144}Nd$) estimates (expressed as $\varepsilon_{Nd}$) from fossilized bio-phosphates (fossil fish teeth and/or -debris) and Fe-Mn-encrusted planktic foraminifera[43] that serve as indicators of deep water provenance-/mixing changes, and hence deep-ocean circulation dynamics at IODP Site U1540. Paired $\varepsilon_{Nd}$ measurements of fossil fish teeth/-debris and planktic foraminifera ($n = 3$) match within 2σ-uncertainties of $\varepsilon_{Nd} = 0.3$. The combined $\varepsilon_{Nd}$ record at IODP Site U1540 indicates higher (i.e., more radiogenic) values during MIS10 and 12 ($\varepsilon_{Nd} = -6.3 \pm 0.3$ ($n = 1$) and $\varepsilon_{Nd} = -5.9 \pm 0.5$ ($n = 19$), respectively) than during MIS11 ($\varepsilon_{Nd} = -7.2 \pm 0.3$, $n = 60$; Fig. 3g). In addition, overall low $\varepsilon_{Nd}$ values during MIS11 are interrupted by shifts towards higher $\varepsilon_{Nd}$ values by 0.4–1.1 $\varepsilon_{Nd}$ units, which exceeds the long-term 2σ-external reproducibility of 0.3 $\varepsilon_{Nd}$ units (Fig. 3g). These $\varepsilon_{Nd}$ shifts occur at the same time as the observed increases in *G. bulloides* and *Uvigerina* spp. U/Ca and U/Mn levels during EII and EIII at our study site, while the $\varepsilon_{Nd}$ shift during EI is less pronounced (Fig. 3b, g).

### Ice-rafted detritus supply

The abundance of ice-rafted detritus (IRD) indicates the presence of icebergs at IODP Site U1540, which is a function of their production rate at the Antarctic margin and their transport efficiency to the study site. IRD abundances are highest during MIS12, and low or (near-)zero during MIS11 and 10 (Fig. 3a). During MIS11c and MIS11b, however, short-lived IRD peaks at respectively ~410–408 ka BP and ~394–390 ka BP coincide with elevated *G. bulloides* and *Uvigerina* spp. U/Ca and U/Mn ratios during EI and EIII, respectively (Fig. 3a, b).

### Bottom water temperature changes

*Uvigerina* spp. Mg/Ca ratios were shown to reflect BWT variations[44], and serve here to estimate CDW temperature variations at our study site. To evaluate the bias from contamination on *Uvigerina* spp. Mg/Ca ratios, we measured Al/Ca, Fe/Ca and Mn/Ca ratios on the same samples. *Uvigerina* spp. Al/Ca is below the limit of detection (<4.7 μmol/mol) in all except 4 out of 80 samples and does not show a statistically significant correlation to *Uvigerina* spp. Mg/Ca ratios within 95% confidence levels ($r^2 = 0.15$, $p = 0.61$). *Uvigerina* spp. Fe/Ca is below the

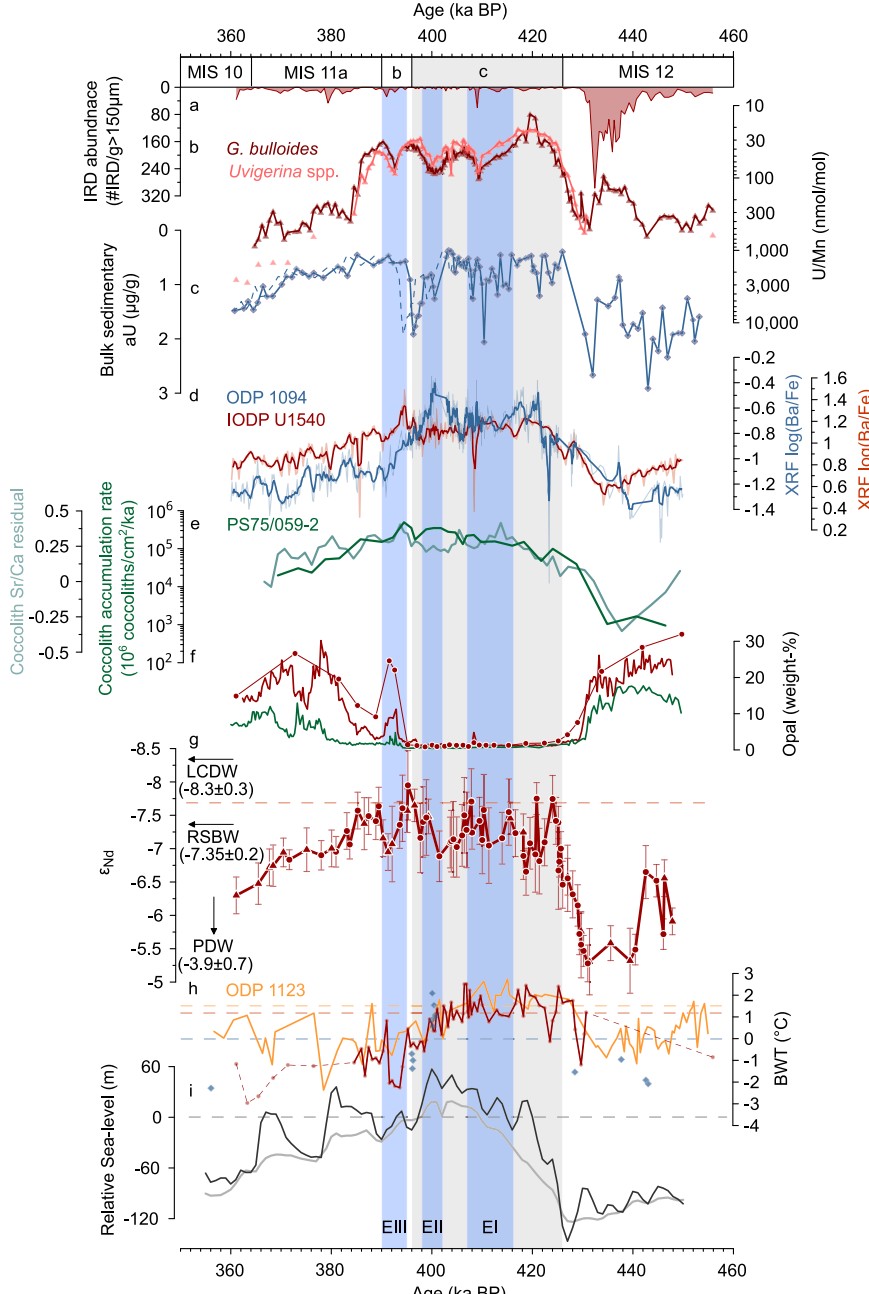

**Fig. 3 | Deep water dynamics and productivity changes in the central Pacific and Atlantic Southern Ocean during Marine Isotope Stage (MIS) 12 to 10.** **a** Abundances of ice-rafted detritus (IRD) at IODP Site U1540. **b** *G. bulloides* (dark red) and *Uvigerina* spp. (light red) U/Mn ratios at IODP Site U1540. **c** Bulk sedimentary authigenic U (aU) levels[26] at ODP Site 1094. **d** X-ray fluorescence (XRF) log(Ba/Fe) ratios at IODP Site U1540 (red) and ODP Site 1094 (blue)[39]. **e** Coccolith accumulation rate (dark green) and temperature-corrected ("residual") coccolith Sr/Ca variations (light green) in core PS75/059-2 in the central South Pacific Ocean[41]. **f** Biogenic opal content in PS75/059-2 (green)[40] and IODP Site U1540 (red)[42]. Symbols indicate discrete opal measurements at IODP Site U1540. **g** Nd isotopic composition ($\varepsilon_{Nd}$) of mixed planktic foraminifera (circles) and fossil fish teeth/-debris (triangles) at IODP Site U1540 (red) including their external (2σ) or internal (2 SE) error. Dashed line and arrows indicate modern $\varepsilon_{Nd}$ composition at IODP Site U1540 (–7.7)[55] and of Ross Sea Bottom Water at hydrographic stations closest to the Ross Sea (i.e., PS75/088 and PS75/089)[54] (RSBW; −7.35 ± 0.2), Lower Circumpolar Deep Water (LCDW, −8.3 ± 0.3)[56] and Pacific Deep Water (PDW, −3.9 ± 0.7)[53], respectively. **h** Bottom water temperature (BWT) estimates at IODP Site U1540 (red), ODP Site 1123 (yellow)[45] and ODP Site 1094 (blue)[36]. Dashed lines indicate present-day BWT (IODP Site U1540 (red): -1.1 °C, ODP Site 1123 (yellow): -1.5 °C, ODP Site 1094 (blue): -0 °C)[46]. **i** Global sea-level variations reconstructed from ODP Site 1123 (black)[45] and a global compilation (gray)[68] (dashed line indicates present-day global mean sea-level). MIS and substages are indicated following ref. [85], with MIS11c indicated by the gray box[28]. Transient peaks in **b** foraminiferal U/Mn levels at IODP Site U1540 during MIS11 are highlighted with blue boxes labeled with roman numbers, i.e., deoxygenation events (E) EI-EIII.

limit of detection (<2.1 µmol/mol) in all except 14 samples and does not correlate with *Uvigerina* spp. Mg/Ca values ($r^2 = 0.04$, $p = 0.47$). *Uvigerina* spp. Mn/Ca values vary between 50 and 220 µmol/mol and show a statistically significant correlation with Mg/Ca values ($r^2 = 0.56$, $p = 0.0001$), likely indicating climatically driven changes in the

precipitation of Mn-oxyhydroxide in marine subsurface sediments rather than a significant bias affecting the Mg/Ca-based BWT estimates at our study site[45]. Indeed, assuming a Mg/Mn ratio in authigenic coatings[45] of 0.1 mol/mol found in oceanic ferromanganese crusts, contaminant Mg/Ca would contribute a maximum of 0.02 mmol/mol

to overall Mg/Ca values and would affect the BWT results insignificantly (<0.2 °C). The combined evidence suggests that the impact of contamination on reconstructed BWT based on *Uvigerina* spp. Mg/Ca ratios at IODP Site U1540 is negligible.

While *Uvigerina* spp. Mg/Ca-derived BWT approach the freezing point during MIS10 and 12, they reach present-day annual-mean levels (-1.1 °C)[46] during the climatic optimum of MIS11 (MIS11c: 1.1 ± 0.8 °C, $n = 50$; Fig. 3h). In addition, during phases of MIS11c, i.e., ~426–416 ka BP and ~410–406 ka BP, BWT at IODP Site U1540 exceed present-day levels by ~1 °C, before they progressively decrease to MIS10 levels, with a strong BWT minimum at ~−2 °C during MIS11b, i.e., ~396–390 ka BP (Fig. 3h). Periods with elevated foraminiferal U/Ca and U/Mn ratios at our study site are therefore characterized by BWT exceeding present-day levels (EI), declining BWT (EII) and BWT levels near freezing (EIII; Fig. 3h). This is confirmed by similar BWT estimates at deep Southeast Pacific[45] ODP Site 1123 and Atlantic Southern Ocean[36] ODP Site 1094 (Fig. 3h).

## Discussion

Observed episodic aU enrichments in foraminiferal coatings at IODP Site U1540 during MIS11 hint at millennial-scale reductions in dissolved oxygen levels in porewaters and by inference in BWO levels (Fig. 2b, c). Paleoproductivity reconstructions at IODP Site U1540 and in the nearby core[41] PS75/059-2 do not show a consistent increase in export production concomitant to the observed foraminiferal U/Ca- and U/Mn peaks (Fig. 3d–f). Along with parallel changes in water mass provenance/-mixing derived from $\varepsilon_{Nd}$ (Fig. 3g) and our BWO estimates based on foraminiferal U/Ca and U/Mn ratios at our study site (Fig. 3b), we exclude a primary control of export production and the associated decomposition of organic matter at depth on our porewater and BWO reconstructions. Additionally, we consider changes in sedimentation rates and effects from remobilization of aU (i.e., "re-burn") to be negligible given the high sedimentation rates during MIS11c at our study site (~9.3 cm/ka) that exceed suggested thresholds indicating a significant impact from re-burn[47]. We therefore argue that biological processes alone did not control reconstructed BWO levels in the central South Pacific Ocean during MIS11. Instead, changes in the advection, formation rate and/or oxygenation of AABW, which supplies $O_2$ to the deep Southern Ocean today[48] (Fig. 1), need to be considered to explain millennial-scale BWO depletions at IODP Site U1540 during MIS11. Our data thereby provide evidence for the presence of 'AABW deoxygenation events' in the Pacific Southern Ocean, reminiscent of the AABW stagnation[27] event identified in the Atlantic Southern Ocean during MIS5e.

The timing of the foraminiferal-derived aU enrichments at IODP Site U1540 during MIS11 is, within age uncertainties, strikingly similar to variations in bulk sedimentary aU enrichment[26] at ODP Site 1094 (Fig. 2d). Two of these aU peaks during MIS11c, i.e., at ~416–408 ka BP and ~403–400 ka BP, in the Atlantic Southern Ocean were suggested to coincide with increased export production[26]. However, we observe similar peaks in aU enrichment in foraminifera in the central South Pacific Ocean (i.e., EI and EII) which we argue to be independent from export production. The synchrony of these BWO reductions in the Atlantic and Pacific sectors of the Southern Ocean implies a common physical-dynamical control via perturbation of AABW formation and/or -advection to both sites simultaneously (Fig. 3). Additionally, considering large age uncertainties during MIS11b at ODP Site 1094 (Fig. 2f; Results), we further argue that bottom water deoxygenation at both sites during MIS11b (i.e., EIII) may also indicate a common decline in the physical/dynamical supply of $O_2$ to bottom waters and not increased export production although a temporal offset cannot be excluded (Fig. 3). Therefore, we argue that aU-derived deoxygenation events observed at IODP Site U1540 and ODP Site 1094 during MIS11c (and possibly MIS11b) suggest wide-spread and simultaneous AABW perturbation, affecting the Pacific and Atlantic sectors of the Southern

Ocean during the protracted, excess warm interglacial of MIS11; yet, they must have originated from different regions around the Antarctic margin.

AABW stagnation events observed in the Atlantic Southern Ocean were linked with Antarctic ice melt-induced surface water freshening in the Weddell Sea Embayment, hence curbing the formation of $O_2$-rich AABW via the suppression of DSW formation[26,27] and its supply into abyssal ocean and Lower CDW. Our data emphasize that it is likely that both the Ross Sea and the Weddell Sea could have been affected by significant meltwater supply due to West Antarctic ice sheet/shelf destabilization and a surface freshening-induced density stratification of the upper water column, driving a synchronous decline in AABW formation and/or AABW $O_2$ levels. This is supported by numerical model simulations that show a synchronous response of WAIS retreat in the Ross Sea-, Weddell Sea- and Amundsen Sea sectors in response to oceanic and atmospheric warming[33]. Destabilization of WAIS is promoted by a poleward shift and/or strengthening of the southern hemisphere westerly winds and/or weakening of the West Antarctic easterlies, favoring increased open-ocean heat (i.e., CDW) supply to the WAIS margin[19,49–51], and hampering DSW- and therefore AABW production[52]. Our proxy results highlight the synchronous occurrence of AABW deoxygenation- (and stagnation-) events in the Pacific and Atlantic Southern Ocean suggesting WAIS destabilization in the Weddell- and Ross Sea Embayment during MIS11c and 11b, likely with a significant role of increased ocean heat exposure of ice shelves in these regions during MIS11c (i.e., EI and EII), which we discuss in detail below.

Additional proxy-evidence from IODP Site U1540 support our inference of a WAIS retreat-driven meltwater supply to the Southern Ocean that curbed AABW formation in the Ross Sea and hence decelerated and/or stagnated AABW flow. First, the short-term and small, yet significant, abundance peaks of IRD at IODP Site U1540 during two of the identified AABW deoxygenation events in MIS11c and MIS11b (i.e., EI and EIII) hint at enhanced iceberg calving (Fig. 3a). This may suggest WAIS retreat and meltwater supply to the Southern Ocean at those times, especially in MIS11c (i.e., EI) because enhanced iceberg survivability due to colder SSTs (Fig. 2e) as a main driver of the observed IRD peak can be excluded. Second, the observed BWO depletion events at our study site are synchronous with marked variations in reconstructed seawater $\varepsilon_{Nd}$ (Fig. 3g). North Atlantic Deep Water (NADW) has very low $\varepsilon_{Nd}$ values[53] (−13.5 ± 0.4), while Pacific Deep Water (PDW) is characterized by more radiogenic $\varepsilon_{Nd}$ values[53] (−3.9 ± 0.7). Lower CDW is mainly influenced by NADW and PDW but also by Ross Sea-derived AABW (or Ross Sea Bottom Water, RSBW) from below. RSBW shows $\varepsilon_{Nd}$ values of −7.35 ± 0.2 in the deep Pacific sector of the Southern Ocean closest to its source in the Ross Sea[54]. Lower CDW is therefore characterized by intermediate $\varepsilon_{Nd}$ signatures[55,56] of −8.3 ± 0.3. Changes in mixing proportions of these constituent water masses are reflected in $\varepsilon_{Nd}$ shifts at our study site. While our $\varepsilon_{Nd}$ record during MIS11 matches Holocene $\varepsilon_{Nd}$ estimates in core PS75/056-1 from the same site[55] (~−7.7), we observe an increase of $\varepsilon_{Nd}$ values by 0.4–1.1 $\varepsilon_{Nd}$ units to $\varepsilon_{Nd}$ values of −7.0 ± 0.4 coinciding with the identified AABW deoxygenation events at our study site (Fig. 3g). This is most pronounced during EII and EIII (Fig. 3g). We interpret these episodic shifts as a reduced influence from NADW and/or Ross Sea-derived AABW (i.e., RSBW), leading to a greater contribution from PDW to Lower CDW at our study site. As NADW flux variations[57] during MIS11 do not coincide with millennial-scale $\varepsilon_{Nd}$ changes at our study site, we argue that instead a perturbation of AABW formation in the Ross Sea Embayment occurred during the identified AABW deoxygenation events at IODP Site U1540 during MIS11.

Numerical models demonstrate a positive feedback mechanism between WAIS melt and hydrographic changes in the Southern Ocean, where meltwater-induced freshening accelerates ice sheet/shelf melting through poleward shifted CDW and/or enhanced CDW intrusion

onto Antarctic continental shelves, potentially triggering a runaway effect of WAIS retreat[16–18,24]. While in principle our proxy data aligns with model predictions of AABW deoxygenation/contraction due to WAIS retreat-driven meltwater supply, they also highlight negative feedbacks promoting millennial-scale reoxygenation of AABW that appear to be not well represented in numerical model simulations. For instance, WAIS regrowth during MIS11 on timescales of millennia might be promoted by glacio-isostatic WAIS uplift[58,59] and/or a significantly reduced sensitivity to ocean warming given the absence or strong spatial reduction of the ice shelves. Alternatively, strong meltwater supply into the Southern Ocean might have strengthened the density gradient along the shelf break between the interior open ocean and the shelf environment[60], protecting the WAIS margin from open ocean heat and/or may have promoted sea ice formation that led to a northward shift and/or weakening of the southern hemisphere westerly winds. Both processes may have suppressed and/or moved the CDW upwelling region away from the Antarctic margin, reducing WAIS exposure to CDW heat[19] and temporarily stabilizing the WAIS margins. Expanded Antarctic sea ice due to SST decline (i.e., following EI; Fig. 2e) and a fresher ocean surface (i.e., during meltwater supply) and northward shifted westerly winds could also have promoted the development of an open-ocean polynya favoring deep-reaching convection[61] (in contrast to coastal polynyas common today), thereby possibly contributing to drive recurring cycles of de-/re-oxygenation and contraction/expansion of AABW observed at our study site during MIS11. Mechanisms driving these cycles, however, need to be further tested with additional proxy data and/or numerical model simulations.

Two of our three identified AABW deoxygenation events at IODP Site U1540 (i.e., EI and EII during MIS11c) can be linked to warm and warmer-than-present Lower CDW as derived from *Uvigerina* spp. Mg/Ca-based BWT estimates at our study site (Fig. 3h). Today, the eastern Ross Sea is mostly a fresh shelf environment with a strong density gradient and isopycnals that tilt towards the continental slope, acting to largely separate cold and fresh shelf waters from warm and salty CDW offshore[20]. The western Ross Sea today is characterized by a dense shelf, where warm CDW is mostly confined to the continental shelf break due to strong incropping isopycnals, allowing DSW formation in coastal polynyas and beneath the Ross Sea ice shelves, suppressing the entrainment of warm CDW onto the continental shelf[20]. The strong isopycnal gradient on the Antarctic slope today with intense AABW formation in the Ross Sea might provide parallels to climate background conditions of MIS11c (Fig. 4a). However, during MIS11c our combined Lower CDW temperature and $O_2$ reconstructions argue for millennial-scale (excess) warm CDW intrusions onto the continental shelf of the Ross Sea Embayment, presumably along the eastern limb of the Ross Gyre – a major conduit of ocean (i.e., CDW) heat between the open ocean and the West Antarctic margin today[23,62] (Fig. 4b, c). Enhanced CDW upwelling and the erosion of isopycnal gradients on the continental slope of Ross Sea (Fig. 4b) would have turned the dense water-shelf regimes into a warm water-shelf environment with accompanying WAIS melt-driven AABW perturbation via meltwater supply[6,9,12,16,18] (Fig. 4b, c).

Rather than driven by CDW heat supply into the Ross Sea Embayment directly, AABW deoxygenation (and by inference stagnation) events observed at IODP Site U1540 during MIS11c might have also resulted from upstream CDW intrusions onto the continental shelf of the Amundsen- and Bellingshausen Sea (Fig. 4d)[23,63]. Today, a strong isopycnal barrier to offshore CDW in the Amundsen- and Bellingshausen Sea is absent[20], allowing warm, 'undiluted' CDW to enter the continental shelf causing high basal WAIS melt rates and creating a warm water cavity[20,21]. An intensification of these warm water-shelf conditions, likely promoted by an expansion of the Ross Gyre as highlighted by numerical Lagrangian particle release experiments[23], may have caused significant melting of ice shelves and the hinterland

ice sheet in that region. Discharged meltwater was then likely advected by the clockwise circulating Ross Gyre towards the Ross Sea[23,63] impeding DSW production[8,63], and subsequently suppressing the formation and dispersal of AABW in the deep South Pacific[23,25] (Fig. 4d). The high sensitivity of the Amundsen- and Bellingshausen Sea to CDW warming may thus imply a downstream perturbation in AABW formation, and hence the oxygenation of the deep South Pacific Ocean, during the AABW deoxygenation events in MIS11c (i.e., EI and EII). Whether CDW heat supply to the Ross Sea or the Amundsen- and Bellingshausen Sea, or in fact a combination of both, was more likely to have occurred needs to be further tested with numerical model simulations ideally using coupled atmosphere-ocean-ice sheet models that allow an assessment of WAIS stability as a function of changes in CDW temperature and/or -exposure time as well as the geometry of the Ross Gyre.

During the identified AABW deoxygenation event during MIS11b (i.e., EIII), CDW temperatures strongly decreased, approaching the freezing point of seawater, similar to the near synchronous stagnation event identified in the Atlantic Southern Ocean[26] at ODP Site 1094 that also coincides with low BWT (Fig. 3h)[36]. We argue that given a lowering of global sea-level (Fig. 3i) and the glacial inception, WAIS grounding line likely advanced northward across the Ross Sea shelf (Fig. 4e). The accompanied expansion of the ice shelf and sea ice in the Ross Sea likely acted as a barrier restricting shelf water to exchange $O_2$ with the atmosphere, which would have led to less ventilated AABW. Furthermore, the mode of AABW formation likely shifted because of these changes, i.e., the relative proportion of supercooling under ice shelves increased over brine expulsion in polynyas. This could have also led to a cooling of AABW, and by inference Lower CDW, consistent with observations in the Pacific and Atlantic Southern Ocean[36] (Fig. 3h). Similar processes might have operated in the West Antarctic continental shelf region of the Weddell Sea given the presence of the vast Filchner-Ronne-Ice Shelf, driving bottom water deoxygenation and -cooling in the Atlantic Southern Ocean[26] during MIS11b; however, the timing of this event at ODP Site 1094 relative to EIII at IODP Site U1540 given the large age uncertainties at ODP Site 1094 remains unclear (see Results). A northward shift and/or weakening of southern hemisphere westerlies[26] during MIS11b could have also reduced the contribution of AABW to Lower CDW, and hence declined BWO levels in Lower CDW during EIII. Irrespective of the complex nature of glacial inception following MIS11c, a combination of these processes likely lowered the $O_2$ content in AABW source waters and hence Lower CDW, and/or decreased the rate of AABW formation via a mode change, which then led to a deoxygenation event at our study site (and possibly similarly and/or synchronously in the Atlantic Southern Ocean[26]) during MIS11b (Fig. 4e). Additional high-resolution proxy data for MIS11 or other interglacial periods for instance[26,27] from MIS5e would be helpful to better constrain the occurrence, timing and mechanisms governing such "cold" AABW deoxygenation events in the Southern Ocean.

Overall, WAIS stability is strongly influenced by CDW temperatures and the duration of the exposure of ice shelves and/or grounding lines to this warm water mass[19,64], which may cause retreat within (sub-)millennial timescales[65]. Numerous model studies have assessed ocean temperature thresholds at which significant WAIS retreat is likely to occur, varying from a few tenths[64] of °C to ~4 °C above present-day levels[15,66,67]. Some numerical simulations also highlight the importance of exposure length over warming intensity, where a moderate ocean warming of 0.4 °C can trigger WAIS collapse if the heat exposure is sustained[64] for at least 4 ka. Our proxy data are in line with these numerical model simulations and suggest that sustained, even modest CDW warming of ~1 °C above present-day levels, can lead to WAIS instabilities, priming WAIS to be increasingly sensitive to prolonged CDW exposure at lower temperatures.

Our findings highlight the impact of ocean warming and sustained CDW upwelling onto the Antarctic continental shelf in the Ross Sea on

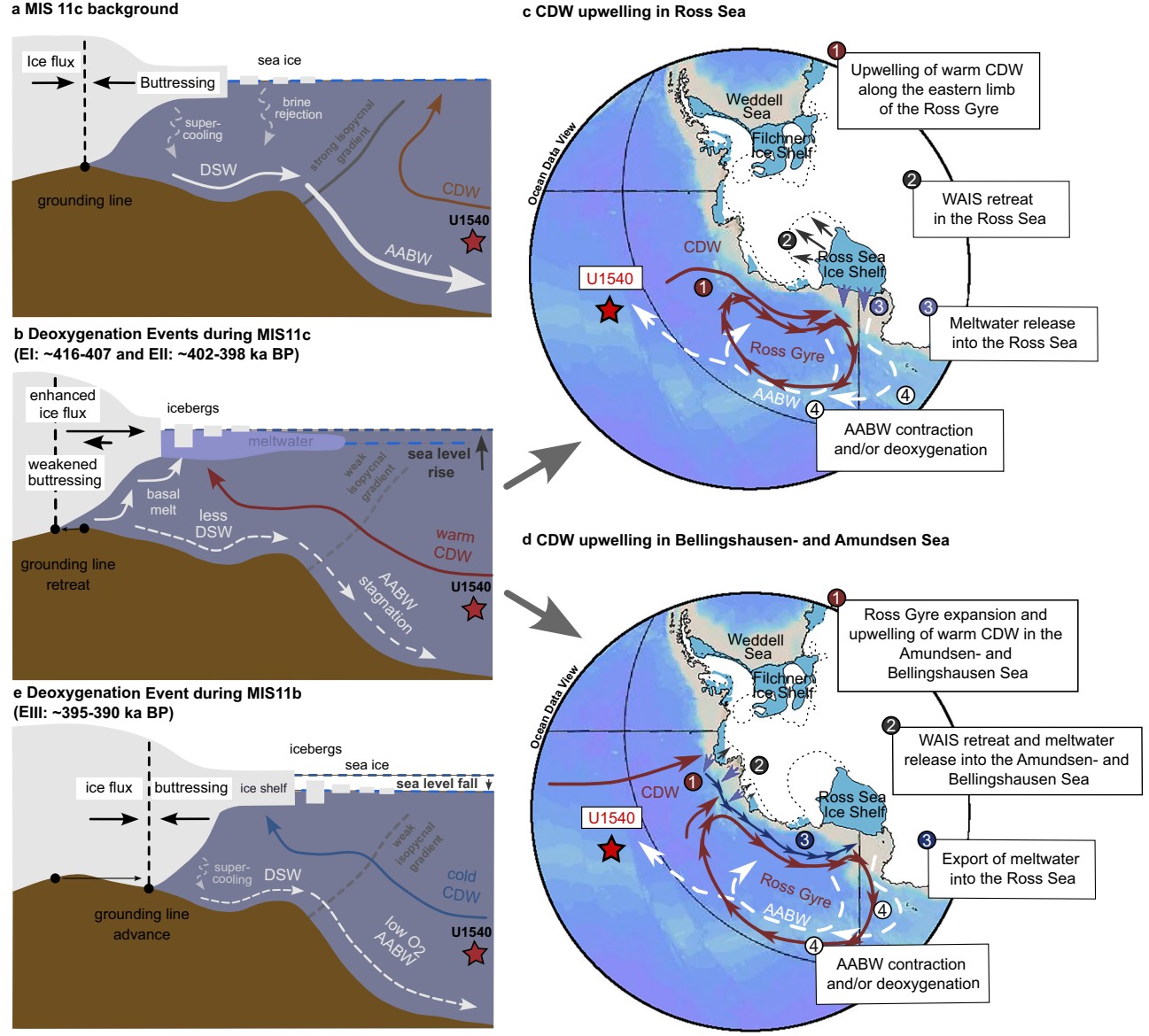

**Fig. 4 | Interplay between West Antarctic ice sheet (WAIS) dynamics and the Southern Ocean during Marine Isotope Stage (MIS) 11 based on proxy analyses at IODP Site U1540 (star). a** Cold water-cavity scenario – background conditions of MIS11c: strong isopycnal gradient at the Antarctic slope of the Ross Sea (gray line) prevents upwelling of Circumpolar Deep Water (CDW) onto the continental shelf (red arrow), leading to minimal basal melting beneath ice shelves and at the WAIS grounding line (dashed black line), favoring Dense Shelf Water (DSW) and Antarctic Bottom Water (AABW) formation (white lines) through supercooling underneath ice shelves and/or brine rejection in coastal polynyas. This leads to well-oxygenated conditions at IODP Site U1540. **b** Warm-water cavity scenario – deoxygenation events (E) EI and EII during MIS11c: isopycnal gradient at the Antarctic slope of the Ross Sea and/or Amundsen/Bellingshausen Seas (gray line) weakens, allowing more and/or warmer CDW to upwell onto the continental shelf. This causes strong basal

melting and WAIS retreat (dashed black line), driving global sea-level rise (dark gray arrow). Two pathways for CDW supply to the Antarctic periphery are possible[23]: to **c** the Ross Sea along Ross Gyre's eastern limb and/or **d** the Amundsen- and Bellingshausen Sea via Ross Gyre expansion (see Discussion and text in figure). **e** Adjusted cold water-cavity scenario – deoxygenation event EIII during MIS11b: Global cooling leads to a northward advance of the WAIS grounding line (dashed black line), which decreases global sea-levels below present-day values (dark gray arrow) and is likely associated with ice shelf- and Antarctic sea ice expansion in the Ross Sea. Cooling of CDW could have changed the rate and/or mode of DSW- and AABW formation, and could have led to cold and low-O₂ AABW, as identified at IODP Site U1540. Possible glacio-isostatic adjustments are neglected here. Maps were created with Ocean Data View[84].

the WAIS geometry/extent, and by consequence on AABW formation during MIS11c (i.e., EI and EII). We therefore suggest significant WAIS retreat beyond present-day grounding lines with substantial ice mass loss during the observed AABW deoxygenation events during MIS11c. Indeed, these events match with peaks in reconstructed global sea-levels (Fig. 3i)[45,68], reaching consensus levels of up to 6–13 m above present-day[32]. Further, during MIS11b (i.e., EIII), the global sea-level reconstruction[45] from ODP Site 1123 suggests a transient sea-level high-

stand of several m (Fig. 3i). Both may indicate indeed perturbations in AABW formation via meltwater supply along the Antarctic margin rather than mechanisms independent from changes in the AABW formation rate[26], and further supports a significant WAIS contribution to global sea-level rise during MIS11c (and possibly MIS11b), as supported by numerical model simulations[15,33]. Our inferences are further strikingly consistent with the timing of WAIS collapse in numerical simulations[64] indicating a WAIS melt-driven global sea-level

contribution of 4.0–8.2 m sea-level equivalent at 412 ka BP – similar to the first AABW deoxygenation event at IODP Site U1540 (i.e., EI). Hence, our findings support ocean-ice sheet models emphasizing the strong sensitivity of WAIS retreat, both in the Ross and Weddell Sea, towards protracted CDW exposure with temperatures close to or slightly above present-day levels rather than CDW warming intensity alone[64].

Our findings of recurring phases of WAIS retreat across ocean basins through the removal of Antarctic ice shelves and accelerated Antarctic ice sheet flow under warmer-than-present and protracted interglacial conditions of MIS11c are reminiscent of observations today, i.e., WAIS mass loss[1], ocean-driven melting of Antarctic ice shelves[2] due to exposure to CDW heat[50], CDW warming[12], AABW perturbations in the Ross Sea[11–13] and an increased open ocean-to-Ross Sea CDW heat transport via the Ross Gyre[23]. Numerical simulations also highlight a significant WAIS contribution to global sea-level rise under unmitigated climate warming in the future[15,17] due to more efficient ocean heat supply to the Antarctic margin[9,16,18]. Our marine sediment core analyses therefore, provide unique observational constraints on the response and sensitivity of WAIS to protracted ocean (i.e., CDW) warming, adjusted pathways of southward ocean heat transport, and the impact of WAIS-melt on AABW formation and deep-ocean $O_2$ supply that are likely to occur in the future with continued ocean warming.

## Methods
### Study area
Sediment cores from IODP Site U1540 (55°08.467'S, 114°50.515'W; 3580 m water depth) were drilled on the eastern flank of the East Pacific Rise north of the modern sub-Antarctic Front in the central South Pacific (Fig. 1)[69]. They contain well-preserved calcareous ooze[69] during MIS11 and siliceous ooze in MIS10 and 12. The high carbonate preservation and high sedimentation rates averaging ~7.5 cm/ka during MIS11 make the core site ideally suited to study the nature, drivers and impacts of bottom water $O_2$ variations on (sub-)millennial timescales and their links to AABW- and Lower CDW dynamics in the central South Pacific Ocean.

There are several distinct sites of AABW formation in the Antarctic periphery with two major sites in the Ross Sea and Weddell Sea located along the WAIS margin[8,25]. The formation of AABW is generally driven by the entrainment of CDW into DSW (i.e., the AABW precursor water mass) that forms on the Antarctic continental shelf and spills over the Antarctic continental margin into the abyssal Southern Ocean due to its high density[8]. Although DSW formation is mostly controlled by brine formation during sea ice formation in polynyas and cooling beneath vast Antarctic ice shelves[52,70,71], spatial differences exist in the formation mode, the hydrographic characteristics and the pathway of AABW (and its precursor water masses) in the abyssal ocean[8,25]. AABW dispersal in the global deep ocean is strongly bathymetry-controlled, with AABW sourced from the Weddell Sea and the Ross Sea occupying mostly the Atlantic sector and Indo-Pacific sector of the Southern Ocean, respectively[8,25] (Fig. 1b). While Lower CDW at our study site, i.e., IODP Site U1540, is primarily influenced by Ross Sea-sourced AABW, ODP Site 1094 is mostly bathed in Lower CDW that is impacted by Weddell Sea-derived AABW (Fig. 1b).

Dense bottom water formation in the Ross Sea accounts for ~25% of AABW formation at present-day[8]. In this region, dense water formation primarily occurs during brine rejection during sea ice formation with only a minor contribution from supercooling of seawater under the Ross Sea Ice Shelf[62]. The western Ross Sea produces the saltiest variety of bottom waters (S > 34.72) with relatively high temperatures of −0.6–-0.3 °C, while bottom waters that originate in the eastern Ross Sea are characterized by low salinities (S < 34.70) and low temperatures (<−0.8 °C)[8]. AABW emerging from the Ross Sea is $O_2$-enriched ([$O_2$] = 210–225 μmol/kg)[48] and spreads into the deep

Southeast Pacific along the East Pacific Rise, into the deep Southwest Pacific east of Campbell Plateau or westward into the Australian-Antarctic Basin (Fig. 1b)[8,25].

The dominant production site of AABW in the global ocean today is the Weddell Sea (~60 %)[8]. Unlike in the Ross Sea, AABW precursor water masses in the Weddell Sea are formed by both brine rejection during sea ice formation in polynyas and supercooling underneath the floating Filchner-Ronne Ice Shelf[8]. Bottom waters derived from the Weddell Sea comprise the coldest (<−1 °C) and freshest (S < 34.64) AABW variety found in the global ocean[8]. Steered by South Atlantic bathymetry, well-oxygenated Weddell Sea-sourced AABW ([$O_2$] > 250 μmol/kg)[8,48] is exported northward into the Argentine Basin and Cape Basin, with only weak dispersal further eastward into the Indian Southern Ocean and no contributions to the central South Pacific (Fig. 1b)[8,25].

### Sampling and sample processing
In total, 169 sediment samples (10 cm³) were taken from the shipboard splice[69] of IODP Site U1540 at 5 cm-increments for MIS10 and 12, and at 2 to 3 cm-increments for MIS11. All samples were freeze-dried and then washed with deionized water over a sieve with a mesh size of 150 μm. The coarse fraction of the sediment was then dried overnight in an oven at a temperature of 40–50 °C. Dry bulk sediments and the coarse fraction of the sediment samples (>150 μm) were weighed with a Sartorius PT310 top loading balance with a precision better than 0.01 g.

### Sedimentary census counts
Planktic foraminiferal assemblages as well as the abundance of rock and mineral fragments (i.e., IRD) were determined based on the coarse fractions (>150 μm) through sedimentary census counts[72]. Specifically, the samples were split with a micro-splitter, were equally distributed on the counting tray and all intact sedimentary grains were counted under a stereomicroscope until a minimum of 300 intact planktic foraminiferal specimens was reached. The percentage of the planktic foraminifer N. pachyderma was then calculated relative to the total number of planktic foraminifera. Neogloboquadrina pachyderma abundances were then converted[73] to summer SST. The presence of IRD was quantified as the number of IRD per dry weight of the coarse fraction (#IRD/g > 150 μm). The 1σ-uncertainty of the census counts was estimated based on 13 replicate counts of the same sample aliquot and is 0.9 % for the abundance of N. pachyderma and 3.4 #IRD/g > 150 μm.

### Stable oxygen isotope analyses of benthic foraminifera
For stable oxygen isotope analyses, two to five specimens of the epibenthic foraminifera Cibicidoides wuellerstorfi and Cibicides kullenbergi larger than 250 μm were handpicked with a brush under the microscope to obtain at least 20–80 μg CaCO₃. Foraminiferal stable oxygen isotopes were measured with a Thermo Fisher Scientific MAT 253 isotope ratio mass spectrometer coupled to a Kiel IV carbonate preparation device at the Leibniz Laboratory for Radiometric Dating and Stable Isotope Research at Kiel University, Germany. The data are expressed in delta-notation ($\delta^{18}O$) with respect to the Vienna Pee Dee Belemnite (VPDB) standard. The analytical precision of the $\delta^{18}O$ analyses is better than 0.08‰ (1σ), as derived from repeated measurements of different internal and international standards (i.e., NBS19, IAEA-603). The 1σ-uncertainty of full replicate C. wuellerstorfi and C. kullenbergi $\delta^{18}O$ analyses amounts to 0.06‰ VPDB for C. kullenbergi (n = 14) and 0.04‰ VPDB for C. wuellerstorfi (n = 3), which is smaller than the analytical uncertainty. The C. wuellerstorfi and C. kullenbergi $\delta^{18}O$ records were adjusted by +0.64‰ VPDB to account for disequilibrium effects[74].

### Trace element analyses of foraminifera
To reconstruct BWT variations at IODP Site U1540 via Uvigerina spp. Mg/Ca ratios and BWO variations via the U and Mn enrichment in

foraminiferal coatings (i.e., U/Ca- and U/Mn ratios, respectively)[34], the planktic foraminifer *G. bulloides* (20–25 specimen; 250–300 μm size fraction) and the benthic foraminifer *Uvigerina* spp. (15–20 specimen; >250 μm size fraction) were handpicked under the microscope to obtain a total weight of 300–500 μg $CaCO_3$. The selected foraminifera were crushed and were oxidatively cleaned following established protocols[75]. In brief, the first oxidative cleaning step includes several milli-Q and ultrapure methanol rinses to remove adhering clay from the sample. The organic matter was oxidatively removed by adding alkali-buffered 1% $H_2O_2$ solution to the samples that were subsequently placed into a hot water bath at ~90–100 °C for 10 min. Then, the samples were rinsed with milli-Q twice and the remaining silicate particles were individually removed from the samples with a fine brush when present. Before the measurement, the samples were weakly acid-leached with ultrapure 0.001 M $HNO_3$ and dissolved for analysis in ultrapure 0.1 M $HNO_3$ to obtain the sample concentrate.

To quantify major element/Ca ratios (i.e., Mg/Ca, U/Ca, Mn/Ca, Fe/Ca and Al/Ca), the samples were initially diluted by adding 25 μl sample concentrate to 275 μl 2% ultrapure $HNO_3$ – the amounts were adjusted if needed to obtain a Ca concentration of 20 μg/g. The samples were then measured on an inductively coupled plasma-optical emission spectrometer (ICP-OES, Spectro ARCOS II) at the Institute of Geosciences at Kiel University, Germany. The results were calibrated with single element standards and drift-corrected based on multiple measurements of ECRM 752-1 (i.e., limestone powder with Mg/Ca = 3.824 mmol/mol)[76]. The external measurement accuracy of the Mg/Ca ratios was determined based on the reference material JCt-1 (giant clam *Tridacna gigas*)[77]. Our measured Mg/Ca ratio of JCt-1 is 1.245 ± 0.011 mmol/mol (1σ, $n = 18$), which matches within uncertainty the reported value (1.289 ± 0.045 mmol/mol)[77]. The analytical uncertainty of *Uvigerina* spp. Mg/Ca ratios is based on four repeated measurements of the same sample solution and is on average 0.005 mmol/mol. *Uvigerina* spp. Mg/Ca ratios were converted to BWT using the calibration of ref. 44 to ensure comparability to the BWT reconstructions at ODP Site 1123 in the Southwest Pacific[45].

To reconstruct BWO variations at IODP Site U1540, we measured U/Ca- and U/Mn ratios of both the planktic foraminifer *G. bulloides* and the benthic foraminifer *Uvigerina* spp. The foraminiferal U/Ca- and U/Mn ratios reflect the post-depositional, redox-driven authigenic precipitation of U in the sediment, which also occurs around deposited foraminiferal tests, under depleted $O_2$ conditions in bottom waters and marine sub-surface sediments[34]. Because the U concentration in the authigenic coatings of foraminifera (up to 800 nmol/mol) is much higher than the lattice-bound U concentrations of *G. bulloides* and benthic foraminifera (0–20 nmol/mol)[35], the measured U/Ca ratio of weakly or oxidatively cleaned foraminifera primarily indicates the redox-driven U concentration in the authigenic coatings[34]. A normalization of *G. bulloides* and *Uvigerina* spp. U/Ca ratios with Mn/Ca reduces test-morphological biases of measured U/Ca levels[34].

To estimate *G. bulloides* and *Uvigerina* spp. U/Ca and U/Mn levels, the sample concentrates were diluted based on ICP-OES-measured Ca levels to a Ca concentration of 20 μg/g. *G. bulloides* and *Uvigerina* spp. U/Ca and Mn/Ca levels were then measured on an Agilent 7900 inductively coupled plasma-mass spectrometer (ICP-MS) at the Institute of Geosciences at Kiel University, Germany. The drift correction of the samples is based on repeated measurements (every tenth sample) of the JCt-1 standard[77]. The analytical uncertainty of the measurements is based on the relative standard deviation of the signal ratios and is on average ~3.6% for U/Ca and ~1.2% for Mn/Ca. The precision of the U/Ca analyses is based on repeat measurements of the reference material KCp-1 (U/Ca = 1031 ± 139 nmol/mol, 1σ, $n = 13$, fossil coral)[78] and HST 210 (U/Ca = 228 ± 27 nmol/mol, 1σ, $n = 11$, *Globigerina* mud from Sulu Sea, Institute of Geosciences, Kiel).

## Nd isotope reconstructions of seawater

To reconstruct the Nd isotopic composition of seawater as a water provenance indicator, 22 samples of fossilized bio-phosphate (fossil fish teeth and/or -debris) and 58 samples of Fe-Mn-encrusted mixed planktic/mono-specific foraminifera (primarily *Globorotalia inflata*) were picked from the coarse fraction (>150 μm) of IODP Site U1540 sediments (at least 2–3 fish teeth or 5–10 pieces of fish debris and ~40 mg foraminiferal $CaCO_3$). The selected sample material was subsequently cleaned and prepared for analysis following existing protocols[55]. Briefly, planktic foraminiferal samples were gently crushed to open all chambers. Subsequent physical cleaning of detrital particles involved multiple rinses and ultrasonication in Optima grade methanol, followed by multiple rinses in deionized water to ensure any attached clay particles were removed. Cleaned foraminiferal fragments were assessed under the microscope for any remaining clays or extraneous detrital material.

After cleaning, the foraminiferal samples were dissolved in 5% Optima Acetic Acid and fish teeth/-debris samples were dissolved in Optima Aqua regia (1:3 mixture of $HNO_3$ and HCl). Subsequently, the samples were transferred into Teflon beakers and dried down on a hot plate. Nd was isolated by two-step column chemistry to avoid interferences with other elements. For the rare earth element (REE) separation, 100 μl shrink-fit Teflon columns were used that are loaded with Eichrom TRU-spec resin (100–150 μm mesh). A procedural blank was included with each TRU-spec batch. The eluted sample containing only REEs was then dried down and ready for Nd separation. Calibrated Teflon columns pre-loaded with Eichrom LN-spec resin (50–100 μm mesh) were used to isolate Nd from the other REEs. Subsequently, collected Nd fractions were dried down, followed by the removal of any potential organic compounds that could have leaked from the columns using a concentrated 1:3:1 Optima grade mixture of $HNO_3$, HCl and $H_2O_2$. Finally, the samples were dried down and redissolved in 2% Optima grade $HNO_3$ for analysis.

All reported Nd isotope data were analyzed at Pennsylvania State University, University Park, PA, USA, using a Thermo Fischer Scientific Neptune-Plus multicollector-ICP-MS (MC-ICP-MS). A desolvation introduction system (an Apex nebulizer) was used for sample introduction into the MC-ICP-MS. Measured $^{143}Nd/^{144}Nd$ ratios were corrected using a reference value of $^{146}Nd/^{144}Nd = 0.7219$ and an exponential mass fractionation law. Mean $^{143}Nd/^{144}Nd$ ratios of repeated measurements of the international standard JNdi-1 for a given analytical session and the published $^{143}Nd/^{144}Nd$ of JNdi-1 (0.512115)[79] were used to report normalized $^{143}Nd/^{144}Nd$ ratios of the samples. Nd isotope results are reported in ε-notation ($\varepsilon_{Nd}$) as parts per 10,000 deviations from Chondritic Uniform Reservoir levels[80] of $^{143}Nd/^{144}Nd = 0.512638$. The reproducibility of the $\varepsilon_{Nd}$ analyses was estimated based on repeated measurements of the international standard JNdi-1, alongside the samples, and is 0.3 $\varepsilon_{Nd}$-units (2σ).

## X-ray fluorescence scanning and sedimentary opal content

Non-destructive XRF scanning was conducted on the shipboard splice of IODP Site U1540 with 2 cm-resolution using an Avaatech XRF core scanner at Texas A&M University, College Station, TX, USA. Split core surfaces of the archive halves were scraped cleaned and covered with a 4 μm-thin SPEXCertiPrep Ultralene foil to avoid cross-contamination and were then subjected to XRF scanning during consecutive 10 kV-, 30 kV- and 50 kV runs. We used the Ba and Fe intensities (area counts) from the 50 kV-run to calculate the logarithmic ratio of log(Ba/Fe). The sedimentary opal content was measured on 30 samples through wet-chemical digestion and spectrophotometry following established methods[81]. Repeated measurements of the same sample yield an analytical uncertainty of ~0.11 weight-% opal (1σ, $n = 13$). The analysis of internal standards shows a long-term reproducibility of 0.47 weight-% opal (1σ). Our discrete opal measurements are plotted on the high-resolution opal record from IODP Site U1540 that was obtained using

the polynomial regression from core PS75/56-02 from the same study location, which itself was calibrated with the gamma ray attenuation record[69] and discrete opal measurements[42].

## Chronology

The age model for the interval of interest at IODP Site U1540 was established by a stratigraphic alignment of our high-resolution benthic foraminiferal (*C. wuellerstorfi* and *C. kullenbergi*) $\delta^{18}O$ record to the benthic foraminiferal LR04 $\delta^{18}O$ stack[37]. The chronology is based on a linear interpolation between the resulting seven tie points. Resulting sedimentation rates vary between 6–7 cm/ka during MIS10 and 12, and increase up to 9.3 cm/ka during MIS11 (Fig. 2a). Our age model results in an excellent agreement of changes in *N. pachyderma* abundance-based summer SST at the study site with EPICA Dome C (EDC) $\delta D$ variations[30] based on the AICC2023 ice age scale[38] (Fig. 2a, e).

## Data availability

All shipboard data from IODP Expedition 383, including for Site U1540, are available from the IODP data archive (https://web.iodp.tamu.edu/LORE/) and the scientific proceedings of IODP Expedition 383 (ref.[69]). New data presented here are accessible via the PANGAEA database (https://doi.pangaea.de/10.1594/PANGAEA.974158)[82].

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

## Acknowledgements

We sincerely thank the scientific party, technical staff, captain and crew of IODP Expedition 383 (Dynamics of Pacific Antarctic Circumpolar Current, DYNAPACC), who made retrieval of the sediment cores studied here possible. Research samples from IODP were obtained with the help of the local staff at Texas A&M University, College Station, TX, USA. We acknowledge financial support via a European Consortium of Ocean Research Drilling (ECORD) Research Grant to L. Jebasinski. We are also grateful to Karen Bremer and Ulrike Westernströer of the Inorganic Geochemistry Laboratory at Kiel University for help with trace element measurements, Andrea Groth and Luisa Franzen for help with sample preparation, selection of foraminifera and discrete opal measurements. Our gratitude also goes to Dr. Aviv Solodoch for providing model output data of tracer simulations in the Southern Ocean that were used in Fig. 1b. We also thank Dr. Nils Andersen and the technical staff of the Leibniz Laboratory for Radiometric Dating and Stable Isotope Research at Kiel University for conducting foraminiferal stable isotope measurements. Partial analytical funding for this project comes from the US National Science Foundation grant (OCE 2406582) to C. Basak. We also acknowledge the IODP US Science Support Program for funding post-expedition activities via a post-expedition activity award to J. Gottschalk and C. Basak.

## Author contributions

L.J. and J.G. designed the study. G.W., F.L., C.B., M.S.-P., and J.G. collected the core material. L.J. performed census counts and selected the foraminifera and fish remains in the samples for geochemical analyses. L.J. and D.A.F. performed the trace metal analyses. LJ., A.K.I.U.K., and C.B. performed the $\varepsilon_{Nd}$ analyses. L.J. and J.G. analyzed the proxy data and wrote the manuscript with contributions from D.A.F., A.K.I.U.K., C.B., M.S.-P., G.W., and F.L.

## Funding

## Competing interests

The authors declare no competing interests.
