## [Transparent Peer Review file · Nature Communications]

Southern Ocean Evidence for Recurring West Antarctic Ice Sheet Destabilization During Marine Isotope Stage 11

Corresponding Author: Ms Lena Jébasinski

Version 0:

Reviewer comments:

Reviewer #1

(Remarks to the Author)

Review of Jébasinski et al.

This is an impressive study of deep sea oxygenation changes in the Southern Ocean during a past warm climate in the ice age era. The authors have multiple proxies to draw from to conclude that there were millennial-scale low oxygen events in the deep Pacific, which tied together with published results from the deep Atlantic, point to circum-Antarctic ocean response to ice sheet melting. Authigenic uranium, neodymium isotopes, bottom water temperatures, barium and opal content provide a very careful analysis of the suggested events. Honestly, I find the study to be technically impeccable and recommend this be published with a wide audience to incorporate these types of events into climate modeling for the future.

To go further, I can offer only one or two reflective comments. The linkage between synchronous changes in both Ross Sea and Wedell Sea bottom water formation is very interesting. Could the authors suggest a possible test for future evaluation of whether the “c” or “d” scenarios (from Figure 4) for E1 and E11 would have been more likely to have occurred?

I also see the difference in the Events I and II versus III, where the bottom water temperature drop during E11 suggests a “cold” stagnation event, versus the “warm” melt-driven events E1 and E11. Are there any other interglacial periods where there is similar bottom water temperature data available as possible other examples of these “cold” stagnation events?

Final thought regarding the interpretation of the epsilon Nd data. Line 222. “These shifts occur mostly at the same time as the observed MIS11 increases in ...U/Ca...”. Could this assessment be tightened up more statistically or specifically? From my reading of Figure 3, it seems like E11 and E11 have clear increases in eNd but E1 could be arguable.

Reviewer #2

(Remarks to the Author)

Reviewer #3

(Remarks to the Author)

This manuscript presents a new multi-proxy and high time-resolution assessment of the impact of past West Antarctic Ice Sheet (WAIS) retreat events via their impact on Antarctic Bottom Water formation as recorded by bottom water oxygen (BWO) concentrations. Previously, only one site in the Atlantic sector of the Southern Ocean has revealed intervals of low BWO, linked to WAIS retreat and the impact of a warmer ocean. Here, the authors move to the Pacific sector of the Southern Ocean to test these hypothesized processes and feedbacks in another region, but one which remains influenced by WAIS behavior. The study targets a particularly long interglacial (MIS 11) which has many relevant properties for us to consider in projecting future climate changes (outlined in detail, and clearly, in the manuscript). Of importance is the new finding that these low BWO events are also found in MIS 11 in the Pacific sector, revealing that there was a dynamic WAIS here which also impacted AABW formation and properties. Because this also has implications for our sea-level budget contributions from WAIS in the past (and future), the results of this work will be of interest to several audiences: paleoceanographers, geochemists, climate and ice sheet modelers, and glaciologists.

The manuscript is well-written and the graphics are clear. The authors do an excellent job in the early pages at setting out

the modern context and concerns, as well as the paleo-environmental evidence for these BWO events in the recent geological past. The authors have amassed a detailed suite of geochemical indicators, at high temporal resolution, and combined the results in a detailed evaluation which ensures that the key processes can be explored and explained. The Discussion is well-written, and balances evaluation and detail with clear interpretations. My comments here reflect a need for increased clarity in a few places, which should ensure that the nuances as well as the highlights of the results are evident to readers; they do not represent any significant flaws to the work being presented.

1) The authors highlight the synchrony or near-synchrony of events between this site (U1540) and the previous record in the Atlantic (ODP 1094). The age model of ODP 1094 is important here, because the new data seems well-constrained with a high resolution tuning to the EPICA ice core. In line 173 the authors note that “a slight shift of the tiepoint at 395.2 kyr bp ... maintains a good match”, but I don’t find the rationale for why this shift was made. It doesn’t seem to have been made with the assumption that the BWO events should be synchronous, especially given the different process driving this event in U1540. The text here seems to imply that the authors didn’t try to align the BWO signals, but a sentence or two is needed to clarify what was done and why.

2) The Discussion on Drivers of AABW stagnation (starting line 323) includes mention of atmospheric and ocean mechanisms which could lead to WAIS destabilisation which includes mention of ocean warming (line 334). As the ocean warming is discussed in a later section I suggest that the authors flag here that this mechanism will be returned to. I initially read this section with some skepticism having inferred that the authors were proposing ocean temperatures as the driver for all three events, when MIS 11b is different. A brief note to clarify this would help future readers.

3) The role of CDW in AABW stagnation events (line 387 onwards) considers the role of ocean temperatures. The first few paragraphs detail the warming effect, whereas a later paragraph highlights that the event in MIS 11b occurs during cooling (and in fact when bottom water temperature reach their minimum). However, the opening sentences don’t note that only 2 of the 3 stagnation events occur with warmer bottom waters. Rather than saying “events...can be partially linked to warm and warmer-than-present...” (line 388-389) it would be more clear if the authors stated that 2 of the 3 events occurred under these conditions.

4) Lines 424-426 describe a relatively locally important process to account for the low BWO event EIII. Does this mean that we should not expect a similar event to be expressed in the Atlantic sector (influenced by the Weddell Sea), or could it account for the temporal offset? With this in mind, the unclear reason for shifting a tie-point in ODP 1094 emerges again.

Minor comments:

Line 55-6: this is a long and complicated sentence. Rather than trying to combine the stabilization and vulnerability in a single sentence, this may be more clear if 2 sentences are used here, one for each scenario.

Line 76-77: unusual expression of dates. If referring to “thousand years ago” then ka can be used (127 ka), rather than 127 kyr bp. The lowercase bp is confusing because capitalized BP is used to refer to radiocarbon dates.

Line 147 needs to define what the “E” is an abbreviation of (event? And were these events defined by the study of ODP 1094?) They are defined in the figure captions, but they are also never again referred to in the text, which uses the ages instead in the Results and Discussion. The authors should consider whether this labelling of “events” is actually necessary.

Line 188: clarify that you are referring to opal in U1540 and not another site.

Line 374: millennia rather than millennials

Version 1:

Reviewer comments:

Reviewer #1

(Remarks to the Author)

The authors have addressed all my concerns in the revised manuscript.

Reviewer #2

(Remarks to the Author)

The authors have adequately addressed my concerns, and I am satisfied with this version of the manuscript. I have no further comments or suggestions.

Reviewer #3

(Remarks to the Author)

The authors have addressed all of the comments I made in my original review, and have made the required changes to the text in a clear way. I have also checked the replies to the other reviewers. Here, there is a range of complexity in the

responses, due to the complicated ocean circulation systems in this region and the uncertainties around available complementary data which could be used to support the interpretations. But I am satisfied that the authors have responded appropriately and included the suggested edits and clarifications suggested by the other two reviewers. All three reviewers emphasise the importance of the data and the well-written and well-presented manuscript.

Response to Reviewers

Manuscript Title: Southern Ocean Evidence for Recurring West Antarctic Ice Sheet Destabilization During Marine Isotope Stage 11

Manuscript number: NCOMMS-25-28167A

We sincerely thank all three Reviewers for their constructive feedback and for pointing out the importance of our study. Below, we provide a detailed, point-by-point response to all raised comments in blue. Line numbers refer to the manuscript file with tracked changes. Italicized text highlights new text implemented in the revised manuscript.

Reviewer #1:

This is an impressive study of deep sea oxygenation changes in the Southern Ocean during a past warm climate in the ice age era. The authors have multiple proxies to draw from to conclude that there were millennial-scale low oxygen events in the deep Pacific, which tied together with published results from the deep Atlantic, point to circum-Antarctic ocean response to ice sheet melting. Authigenic uranium, neodymium isotopes, bottom water temperatures, barium and opal content provide a very careful analysis of the suggested events. Honestly, I find the study to be technically impeccable and recommend this be published with a wide audience to incorporate these types of events into climate modeling for the future.

We thank Reviewer #1 for the constructive feedback and appreciate his/her time and effort in reviewing our manuscript. We are grateful for the positive assessment.

To go further, I can offer only one or two reflective comments. The linkage between synchronous changes in both Ross Sea and Weddell Sea bottom water formation is very interesting. Could the authors suggest a possible test for future evaluation of whether the “c” or “d” scenarios (from Figure 4) for EI and EII would have been more likely to have occurred?

Based on our data, we unfortunately cannot evaluate which scenario, or whether a combination of both, was more likely to have occurred – this in fact goes beyond the scope of our paper, in our view. However, we argue that additional proxy analyses from marine sediment cores on the continental shelf of the Amundsen/Bellingshausen Sea and the Ross Sea, including CDW temperature and -salinity reconstructions might allow an assessment of whether scenario “c” or/and “d” would have been more likely. These efforts would need to be combined with numerical model simulations using coupled atmosphere-ocean-ice sheet models to identify the most vulnerable areas of the West Antarctic Ice Sheet as a function of degree and/or time of exposure to ocean heat and the geometry of the Ross Gyre. We have accordingly added a sentence in the revised manuscript (lines 450-454), stating “*Whether CDW heat supply to the Ross Sea or the Amundsen- and Bellingshausen Sea, or in fact a combination of both, was more likely to have occurred needs to be further tested with numerical model simulations ideally using coupled atmosphere-ocean-ice sheet models that allow an assessment of WAIS stability as a*

function of changes in CDW temperature and/or -exposure time as well as the geometry of the Ross Gyre.”.

I also see the difference in the Events I and II versus III, where the bottom water temperature drop during EIII suggests a “cold” stagnation event, versus the “warm” melt-driven events EI and EII. Are there any other interglacial periods where there is similar bottom water temperature data available as possible other examples of these “cold” stagnation events?

We indeed differentiate EI and EII from EIII on the basis of possibly different mechanisms driving the events (“warm” versus “cold”). In our manuscript, we show bottom water temperature reconstructions from ODP Sites 1123 (Elderfield et al., 2012) and 1094 (Hasenfratz et al., 2019) that are in strong agreement with our BWT record at IODP Site U1540, in support of our distinction of the events (Figure 3h). However, we are not aware of sufficiently high-resolution data of bottom water temperature and -oxygenation from other interglacial periods and from other locations bathed in CDW to put further constraints on these different events. Frankly, we also believe that comparison of our MIS11 data to other interglacial periods goes beyond the scope of the paper, as these likely are characterized by different climate boundary conditions, durations and/or orbital forcing. Nonetheless, we emphasize in the revised manuscript the consistency of our bottom water temperature estimates with above mentioned records for MIS11 (lines 278-279 and 457-459). Furthermore, we now clarify that additional data from other interglacials would be helpful to further analyze the occurrence of “warm” versus “cold” AABW deoxygenation events: *“Additional high-resolution proxy data for MIS11 or other interglacial periods for instance^{26,27} from MIS5e would be helpful to better constrain the occurrence, timing and mechanisms governing such “cold” AABW deoxygenation events in the Southern Ocean.”* (line 479-481).

Final thought regarding the interpretation of the epsilon Nd data. Line 222. “These shifts occur mostly at the same time as the observed MIS11 increases in ...U/Ca...”. Could this assessment be tightened up more statistically or specifically? From my reading of Figure 3, it seems like EIII and EII have clear increases in eNd but EI could be arguable.

One strength of our study is the multi-proxy approach, where we apply the reconstruction of key parameters on *one and the same* sediment core sequence. This erases stratigraphic biases between proxies that would occur when comparing the same or different proxies in different cores. This is now emphasized in the revised manuscript: *“combined reconstructions of [...] – thereby circumventing stratigraphic biases between proxy records”* (line 108-113). Particularly important to our study is the ϵ_{Nd} record as a biologically independent indicator for water mass provenance changes at our study site in the past. We can thus elegantly address the question whether the deoxygenation signal (U/Ca, U/Mn) is influenced by export production and/or changes in water mass circulation. We also now decided to graphically start the vertical bar of EI at 416 ka BP with the onset of elevated authigenic U levels at ODP Site 1094 *and* of foraminiferal U/Ca and U/Mn levels at our study IODP Site U1540 (Figure 2 and 3), which coincides with an increase in ϵ_{Nd} , yet a weak one, at the same site. We therefore now specify in the revised manuscript, that “These ϵ_{Nd} shifts occur at the same time as the observed increases in *G*.

bulloides and *Uvigerina* spp. U/Ca and U/Mn levels during EII and EIII at our study site, while the ϵ_{Nd} shift during EI is less pronounced (Fig. 3b, g).” (lines 238-241) and “we observe an increase of ϵ_{Nd} values by 0.4–1.1 ϵ_{Nd} units to ϵ_{Nd} values of -7.0 ± 0.4 coinciding with the identified AABW deoxygenation events at our study site (Fig. 3g). This is most pronounced during EII and EIII (Fig. 3g).” (lines 379-382). We hope that this sufficiently specifies the alignment between proxy data in our new high-resolution records.

Reviewer #2:

This study uses a multi-proxy approach to identify recurring millennial-scale deoxygenation episodes in AABW in the central Pacific during MIS 11, concurrent with similar events in the Atlantic. These synchronized AABW stagnation events are linked to WAIS retreat driven by subsurface warming from CDW. The results suggest a circumpolar response to ocean heat, with WAIS meltwater contributing significantly to MIS 11 sea-level high stands and potentially to future sea-level rise under continued warming.

The authors present interesting and timely results, highlighting the physical and dynamical roles of WAIS meltwater and CDW in driving recurring AABW deoxygenation events, with relevance to ongoing and future climate change. Overall, I find this to be a well-written and well-structured paper. However, I have several major concerns, particularly regarding the inconsistent proxy patterns across the proposed AABW stagnation events and the interpretation of key physical processes, as detailed below.

We thank Reviewer #2 for evaluating our manuscript and providing such valuable and constructive feedback, which helped to strengthen the quality of our manuscript. We hope that the clarifications in the revised manuscript suffice to address all important points raised.

Major comments

(1) Recurring events: I appreciate the authors’ use of a multi-proxy approach in Figures 2 and 3, which reveals generally good agreement among several variables during MIS 11. However, not all proxies exhibit consistent recurring patterns across the EI–EIII events. For example, summer SST/water isotope (δD) and benthic foraminiferal $\delta^{18}O$ (Fig. 2), as well as BWT and relative sea level (Fig. 3), do not show clearly repeated signals. Could the authors clarify the reasons for these discrepancies?

We appreciate the praise of Reviewer #2 of our multi-proxy approach and the resulting “good agreement” of our data. We believe that this makes our study particularly strong. In contrast to what Reviewer #2 might imply, we do indeed compare in detail our BWO signals with BWT (starting line 415) and global sea-level (starting line 495).

We believe that the absence of millennial-scale variability in the benthic foraminiferal $\delta^{18}O$ at IODP Site U1540 (Fig. 2f) cannot be used to invalidate the findings of consistent changes in reconstructed BWO, IRD input and ϵ_{Nd} -based water mass proportions, because benthic foraminiferal $\delta^{18}O$ reflects changes in *global* ice volume as well as BWT, bottom

water salinity and foraminiferal vital effects. Benthic foraminiferal $\delta^{18}\text{O}$ alone may therefore not be ideal to reconstruct transient *local* Pacific Southern Ocean bottom water deoxygenation events at our study site (for instance through meltwater supply) due to a strong global overprint, although the deconvolution of these processes to extract the magnitude of bottom water salinity changes from benthic foraminiferal $\delta^{18}\text{O}$ at the study site has merit and is planned for a future study.

Furthermore, summer sea surface temperature at our study site and EDC δD are strongly influenced by numerous processes including variations in atmospheric temperatures, mixed layer depths, seasonality, frontal movements, SWW intensity and/or position as well as greenhouse gas forcing (e.g., Timmermann et al., 2014; Vimeux et al., 2002). They are therefore governed by additional and/or different mechanisms than those necessarily driving bottom water deoxygenation events at our study site, which may explain the apparent inconsistency between records.

The link between our reconstructed BWT and our BWO signals is detailed in the discussion, starting in line 415. Our multi-proxy dataset from one core sequence (i.e., without stratigraphic biases) allows us to differentiate between “warm” AABW deoxygenation events (EI and EII) and a “cold” AABW deoxygenation event (EIII). Our BWT reconstructions thus highlight a complex southern high-latitude ocean-cryosphere system that goes beyond a one-to-one correlation, which we address in the study.

Finally, the link between global sea level reconstructions and our BWO signals is described in the manuscript, starting in line 495. We highlight for instance: “Indeed, *these events* [the observed AABW deoxygenation events during MIS11c] match with peaks in reconstructed global sea-levels (Fig. 3i)^{49,55}, reaching consensus levels of up to 6–13 m above present-day³². Further, during MIS11b (i.e., EIII), the global sea-level reconstruction⁴⁹ from ODP Site 1123 suggests a transient sea-level high-stand of several m (Fig. 3i).” (line 500-504). We also compare these findings against model simulations (Mas e Braga et al., 2021).

We strongly believe that these efforts are sufficient to address the association of the vast number of proxies that we present in our comprehensive study, and hope that Reviewer #2 agrees.

In addition, the term “AABW stagnation events” is not clearly defined in the manuscript. At times, two events are mentioned; at other times, three, depending on the interpretation of the IODP site data. I suggest the authors clearly specify how many such events are explicitly discussed and on what criteria this identification is based.

The term “AABW stagnation event” was originally defined by Hayes et al. (2014) based on findings in the Atlantic Southern Ocean, which we clearly outline in the Introduction (line 78-82). In our manuscript, we mostly deliberately use “deoxygenation event” rather than “stagnation event” to use a term free of a mechanistic implication (please also see our response to a comment of Reviewer #3), and to allow for a discussion of the mechanisms driving these events without implying a driver through their designation at the first instance.

We further define the abbreviation “E” to stand for “deoxygenation event”. In the revised manuscript, we are thus able to define our three events (EI-EIII) more clearly: “*We thus refer to the millennial-scale increases in U/Ca and U/Mn levels at IODP Site U1540 at ~416–407, ~402–398 and ~395–390 ka BP as “deoxygenation events” (hereafter referred to as EI, EII and EIII, respectively; Fig. 2) [...]*” (line 165-167). We now address the event(s) more specifically throughout the revised manuscript (by using the abbreviations EI, EII and/or EIII), which we believe enhances the clarity of our study. Examples are “The first two increases in foraminiferal U/Ca and U/Mn levels during MIS11c *during EI and EII [...]*” (line 171-172), “[...] abundance peaks of IRD at IODP Site U1540 during two of the identified AABW *deoxygenation events in MIS11c and MIS11b (i.e., EI and EIII) [...]*” (line 365-366) and “*Two of our three identified AABW deoxygenation events at IODP Site U1540 (i.e., EI and EII during MIS11c) can be linked to [...]*” (line 415-416).

(2) Negative feedbacks between WAIS retreat and AABW contraction: I have a concern regarding the authors’ statement that “meltwater supply into the Southern Ocean might have steepened the isopycnal gradient along the shelf break” (lines 376–377). Whether such steepening occurs likely depends on how effectively the meltwater is retained on the continental shelf. Given the large volume of meltwater input (~50 m sea-level equivalent, as shown in Fig. 3i), a substantial portion would likely have escaped to the open ocean. In that case, the West Antarctic shelf would not remain isolated, and the isopycnal gradient along the shelf break would more likely be flattened rather than steepened.

We must admit that our statement on changes of the isopycnal gradient along the shelf break due to meltwater supply was imprecise, and accordingly thank the reviewer for pointing that out. Rather than “steepening”, we meant to say that “*strong meltwater supply into the Southern Ocean might have strengthened the density gradient along the shelf break between the interior open ocean and the shelf environment⁶⁴, protecting the WAIS margin from open ocean heat,*” (as now revised in lines 399-401), independent from the amount of meltwater input (i.e., ~6-13 m sea-level equivalent; Dutton et al., 2015) or the steepness (i.e., the geographical tilt) of the isopycnals that may have flattened due to the mechanisms that Reviewer #2 describes. For our argument, however, the magnitude of the seawater density gradient across the shelf break matters, i.e., between the shelf and the Southern Ocean interior, which we now better clarify in the revised manuscript.

(3) Development of an open-ocean polynya: I do not agree with the authors’ statement that “expanded Antarctic sea ice and northward shifted westerly winds could also have promoted the development of an open-ocean polynya favoring deep-reaching convection” (lines 381–383). While I understand that this conclusion is based on the assumption of “a northward shift of the westerlies”, it is important to note that this occurred under warming conditions (as indicated by SST in Fig. 2e). In general, open-ocean polynyas are expected to decrease in a warmer climate due to enhanced stratification and reduced sea ice formation.

Our intention is to highlight the possibility of deep-reaching convection initiated by open-ocean polynyas *as one possible mechanism out of many* that may drive recurring cycles of de-/re-oxygenation and contraction/expansion of AABW – an important observation in our view. Although Reviewer #2 is of course right that open-ocean polynyas decline

under warm conditions, we argue that the meltwater supply (that may disperse into the open ocean as Reviewer #2 rightly mentions in his/her comment (2) above) may act to enhance sea ice formation (due to a fresher surface). Furthermore, our sea surface temperature reconstructions and the EDC δD -based Antarctic temperature record indicate *decreasing temperatures* directly after EI (Fig. 2e), which would again promote the formation of Antarctic sea ice. We therefore find that our assertion is not unlikely, and thus decided to keep our argument on the possibility of open-ocean polynyas included in the revised manuscript. We have, however, adjusted our statement to clarify that: “Expanded Antarctic sea ice *due to SST decline (i.e., following EI; Fig. 2e) and a fresher ocean surface (i.e., during meltwater supply)* and northward shifted westerly winds could also have promoted the development of an open-ocean polynya favoring deep-reaching convection⁶⁵ (in contrast to coastal polynyas common today), thereby *possibly contributing to drive* recurring cycles of de-/re-oxygenation and contraction/expansion of AABW observed at our study site during MIS11.” (line 405-411). And we further emphasize: “*Mechanisms driving these cycles, however, need to be further tested with additional proxy data and/or numerical model simulations.*” (line 411-413).

(4) CDW upwelling in Ross Sea (Fig. 4c): I cannot follow the authors’ statement “warm CDW intrusions onto the continental shelf of the Ross Sea Embayment, presumably along the eastern limb of the Ross Gyre” (lines 399–401). Are the authors referring to CDW itself, or to the Antarctic Slope Current (ASC) as the transport mechanism? I do not understand how CDW would move along the Ross Gyre as shown in Fig. 4c.

The flow path of CDW to the Antarctic margin along the eastern limb of the Ross Gyre has been described in detail by Orsi and Wiederwohl (2009) and Prend et al. (2024), and others, see review of Bennetts et al. (2024) and references therein. This is a widely accepted process: “The [Ross] gyre acts as a conduit between the open ocean and the Antarctic margins, carrying CDW poleward to the continental slope on its eastern edge and exporting shelf waters equatorward along its western flank [...]” (Prend et al., 2024) and “The warm CDW then rises along density surfaces as it crosses the ACC and circulates through the subpolar gyres of the Southern Ocean until it eventually sits at depths comparable to the sea floor of the continental shelf.” (Thompson et al., 2018). This is analogous to how CDW was found to enter the West Antarctic continental margin in the Atlantic Southern Ocean, namely through the eastern boundary of the Weddell Gyre (Hellmer et al., 2012; Rintoul et al., 2001; Schröder & Fahrbach, 1999). Ross- and Weddell Gyre circulation thus allows CDW to impinge and/or cross the ASC to arrive at the shelf break of the West Antarctic continental margins of the Ross Sea and Weddell Sea, and we argue that these dynamics may have controlled heat supply to WAIS in the past. We thus keep our postulations in the revised manuscript (Fig. 4c), and specify in the revised manuscript “[...] reconstructions argue for millennial-scale (excess) warm CDW intrusions onto the continental shelf of the Ross Sea Embayment, presumably along the eastern limb of the Ross Gyre – *a major conduit of ocean (i.e., CDW) heat between the open ocean and the West Antarctic margin today*^{23,66} (Fig. 4b, c).” (line 427-430). We believe that this is important because this conduit has likely operated, yet varied in the

past, which as we argue has implications for the heat supply to the Antarctic margin, and ultimately for the ventilation of the deep ocean via AABW.

Minor comments

In Sections 2.1 and 2.2, please refer specifically to the subpanels in Figures 2 and 3 (e.g., Fig. 2a, 2b, Fig. 3a, 3b, etc.) where appropriate. Given the large number of subplots, it is difficult for readers to follow the discussion without clear and direct references to the relevant panels.

Specified, where possible.

Figure 1b: I am not sure it is appropriate to plot tracer concentrations originating from both the Weddell Sea and the Ross Sea on the same figure. The overlapping region within the Weddell Sea makes the results difficult to interpret.

Thank you for this important observation regarding the distribution of the tracer concentration, specifically in the Weddell Sea. The main purpose of Figure 1b is to convey the different preponderances of Weddell Sea- and Ross Sea-derived AABW in the Atlantic (ODP Site 1094) and Pacific sector of the Southern Ocean (IODP Site U1540), respectively. For clarification, we have included the contours highlighting the areas with tracer concentrations larger than 0.02 for both the Weddell Sea- and Ross Sea experiments to indicate that there is no strong overlap of these water masses in the Weddell Sea (according to the analyses of Solodoch et al., 2022, their Figure 1a, c), please also see revised Figure 1b and associated revised caption. We hope that these changes satisfy the valid comment of Reviewer #2.

Figure 2d caption: Change “age model (see Results)” to “age model (see Methods)”.

Our tiepoint adjustment for ODP Site 1094 is described in the Results chapter (line 181-191). Therefore, we retain the phrasing “age model (see Results)” in the caption of Figure 2d to refer the readers to the correct section.

Reviewer #3:

This manuscript presents a new multi-proxy and high time-resolution assessment of the impact of past West Antarctic Ice Sheet (WAIS) retreat events via their impact on Antarctic Bottom Water formation as recorded by bottom water oxygen (BWO) concentrations. Previously, only one site in the Atlantic sector of the Southern Ocean has revealed intervals of low BWO, linked to WAIS retreat and the impact of a warmer ocean. Here, the authors move to the Pacific sector of the Southern Ocean to test these hypothesized processes and feedbacks in another region, but one which remains influenced by WAIS behavior. The study targets a particularly long interglacial (MIS 11) which has many relevant properties for us to consider in projecting future climate changes (outlined in detail, and clearly, in the manuscript). Of importance is the new finding that these low BWO events are also found in MIS 11 in the Pacific sector, revealing that there was a dynamic WAIS here which also impacted AABW formation and properties. Because this also has implications for our sea-level budget contributions from WAIS in the past (and

future), the results of this work will be of interest to several audiences: paleoceanographers, geochemists, climate and ice sheet modelers, and glaciologists.

The manuscript is well-written and the graphics are clear. The authors do an excellent job in the early pages at setting out the modern context and concerns, as well as the paleo-environmental evidence for these BWO events in the recent geological past. The authors have amassed a detailed suite of geochemical indicators, at high temporal resolution, and combined the results in a detailed evaluation which ensures that the key processes can be explored and explained. The Discussion is well-written, and balances evaluation and detail with clear interpretations. My comments here reflect a need for increased clarity in a few places, which should ensure that the nuances as well as the highlights of the results are evident to readers; they do not represent any significant flaws to the work being presented.

We sincerely thank Reviewer #3 for the constructive and valuable feedback on our manuscript.

(1) The authors highlight the synchrony or near-synchrony of events between this site (U1540) and the previous record in the Atlantic (ODP 1094). The age model of ODP 1094 is important here, because the new data seems well-constrained with a high resolution tuning to the EPICA ice core. In line 173 the authors note that “a slight shift of the tiepoint at 395.2 kyr bp ... maintains a good match”, but I don’t find the rationale for why this shift was made. It doesn’t seem to have been made with the assumption that the BWO events should be synchronous, especially given the different process driving this event in U1540. The text here seems to imply that the authors didn’t try to align the BWO signals, but a sentence or two is needed to clarify what was done and why.

We intend our age model adjustment of ODP Site 1094 (as described in the “Results”) to reflect a transparent sensitivity test to assess the timing of EIII at IODP Site U1540 and ODP Site 1094 in more detail, within viable age uncertainties. We thereby deliberately adhere to the original age model premise of Hasenfratz et al. (2019) of a good match between the benthic foraminiferal $\delta^{18}\text{O}$ record of ODP Site 1094 and the LR04 $\delta^{18}\text{O}$ stack by shifting the tiepoint at 395.2 ka BP by 2 ka within uncertainties of the reported age model (Figure 2d, f). Indeed, without strictly forcing an alignment of the BWO signals, we find that the events occur more closely in time during MIS11b (i.e., near EIII), while maintaining a good match of the benthic $\delta^{18}\text{O}$ records. We thus specify in the revised text: *“An adjustment of the existing age model at ODP Site 1094 within its uncertainties and adhering to the original age model premise³⁸ emphasizes ambiguities regarding the relative timing of EIII at IODP Site U1540 and ODP Site 1094 (Fig. 2f). Specifically, considering a slight shift of the tiepoint at 395.2 ka BP at ODP Site 1094 (by ~2 ka) maintaining a good match between the benthic foraminiferal $\delta^{18}\text{O}$ record to the LR04 $\delta^{18}\text{O}$ stack shifts the aU enrichment event at ODP Site 1094 at 397 ka BP by ~2 ka towards younger ages (Fig. 2d, f). This sensitivity test highlights that the foraminiferal U/Ca and U/Mn increase during MIS11b at IODP Site U1540 (i.e., EIII) might have coincided with a similar aU enrichment event at ODP Site 1094 (Fig. 2d). Without high-resolution $\delta^{18}\text{O}$ data and/or additional age control for ODP Site 1094, it can, however, not be excluded*

that EIII represents a local event at IODP Site U1540.” (line 181-191). We hope that these changes and details suffice to adequately address the valid comment of Reviewer #3.

(2) The Discussion on Drivers of AABW stagnation (starting line 323) includes mention of atmospheric and ocean mechanisms which could lead to WAIS destabilisation which includes mention of ocean warming (line 334). As the ocean warming is discussed in a later section I suggest that the authors flag here that this mechanism will be returned to. I initially read this section with some skepticism having inferred that the authors were proposing ocean temperatures as the driver for all three events, when MIS 11b is different. A brief note to clarify this would help future readers.

We agree and now make a distinction between the events and refer to further details later in the text: “Our proxy results highlight the synchronous occurrence of AABW deoxygenation- (*and stagnation-*) events in the Pacific and Atlantic Southern Ocean suggesting WAIS destabilization in the Weddell- and Ross Sea Embayment during MIS11c and 11b, likely *with a significant role of increased ocean heat exposure of ice shelves in these regions during MIS11c (i.e., EI and EII), which we discuss in detail below.*” (line 355-360).

(3) The role of CDW in AABW stagnation events (line 387 onwards) considers the role of ocean temperatures. The first few paragraphs detail the warming effect, whereas a later paragraph highlights that the event in MIS 11b occurs during cooling (and in fact when bottom water temperature reach their minimum). However, the opening sentences don’t note that only 2 of the 3 stagnation events occur with warmer bottom waters. Rather than saying “events...can be partially linked to warm and warmer-than-present...” (line 388-389) it would be more clear if the authors stated that 2 of the 3 events occurred under these conditions.

We agree that the differentiation between EI and EII (i.e., the “warm” events) and EIII (i.e., “cold” event) needs to be made clearer. We adjusted our statement in lines 415-417 accordingly: “*Two of our three identified AABW deoxygenation events at IODP Site U1540 (i.e., EI and EII during MIS11c) can be linked to warm and warmer-than-present Lower CDW [...].*”. Please also refer to our response to comment (1) of Reviewer #2.

(4) Lines 424-426 describe a relatively locally important process to account for the low BWO event EIII. Does this mean that we should not expect a similar event to be expressed in the Atlantic sector (influenced by the Weddell Sea), or could it account for the temporal offset? With this in mind, the unclear reason for shifting a tie-point in ODP 1094 emerges again.

We agree that processes governing EIII are described with a focus on the Ross Sea (given our study site and the robust age constraints we have for IODP Site U1540 based on independent age constraints i.e., SST-EDC δD alignment; Fig. 2e, f). Yet, we did not intend to exclude the possibility that similar processes operated in the Weddell Sea during the same time. In our view, a near-alignment of the BWO signal during EIII in the Pacific- (IODP Site U1540) and Atlantic Southern Ocean (ODP 1094) (lines 187-190) is equally likely as two locally BWO EIII events with a temporal offset (lines 190-191), given the significant age uncertainties of ODP Site 1094 during EIII (Hasenfratz et al., 2019) and

the results of our age model sensitivity test. We believe that highlighting both in the revised manuscript is transparent, and therefore the best way forward.

In the revised manuscript, we thus add: “During the identified AABW deoxygenation event during MIS11b (*i.e.*, EIII), CDW temperatures strongly decreased, approaching the freezing point of seawater, *similar to the near synchronous stagnation event identified in the Atlantic Southern Ocean²⁶ at ODP Site 1094 that also coincides with low BWT (Fig. 3h)³⁸.*” (line 456-459) and “*Similar processes might have operated in the West Antarctic continental shelf region of the Weddell Sea given the presence of the vast Filchner-Ronne-Ice Shelf, driving bottom water deoxygenation and -cooling in the Atlantic Southern Ocean²⁶ during MIS11b; however, the timing of this event at ODP Site 1094 relative to EIII at IODP Site U1540 given the large age uncertainties at ODP Site 1094 remains unclear (see Results).*” (line 467-472). We hope that these additions sufficiently address the important comment of Reviewer #3.

Minor comments:

Line 55-6: this is a long and complicated sentence. Rather than trying to combine the stabilization and vulnerability in a single sentence, this may be more clear if 2 sentences are used here, one for each scenario.

Amended.

Line 76-77: unusual expression of dates. If referring to “thousand years ago” then ka can be used (127 ka), rather than 127 kyr bp. The lowercase bp is confusing because capitalized BP is used to refer to radiocarbon dates.

As per international convention (ISO 31 - Quantities and units, International Organization for Standardization, 1993) we decided to use the term ka (kilo-annum) before present (BP) for both the age and duration of an event. We accordingly changed all “kyr bp” to “ka BP” in the revised manuscript, including text, figures and figure captions.

Line 147 needs to define what the “E” is an abbreviation of (event? And were these events defined by the study of ODP 1094?) They are defined in the figure captions, but they are also never again referred to in the text, which uses the ages instead in the Results and Discussion. The authors should consider whether this labelling of “events” is actually necessary.

We intend the abbreviation “E” to stand for “deoxygenation event” to reduce complexity (*i.e.*, this designation is free of a mechanistic implication) and enhances in our view the readability of our manuscript. Therefore, we maintain the use of this abbreviation in the revised manuscript but apply it more consistently throughout. We also define its use at the beginning of the “Results” section more clearly: “*We thus refer to the millennial-scale increases in U/Ca and U/Mn levels at IODP Site U1540 at ~416–407, ~402–398 and ~395–390 ka BP as “deoxygenation events” (hereafter referred to as EI, EII and EIII, respectively; Fig. 2), in analogy to the definition of “stagnation event” of ref. ²⁷; yet, we deliberately choose to refer to EI-EIII without a mechanistic implication.*” (line 165-169).

Line 188: clarify that you are referring to opal in U1540 and not another site.

Clarified (now line 203).

Line 374: millennia rather than millennials

Amended (now line 397).

Please note that, based on a minor adjustment of the MIS11c time interval to the margins defined in the introduction (line 98), some of the reported mean values in the manuscript have been revised (e.g., line 236 and 271); however, these minor changes do not affect the interpretation or conclusions of this study.

References

- Bennetts, L. G., Shakespeare, C. J., Vreugdenhil, C. A., Foppert, A., Gayen, B., Meyer, A., Morrison, A. K. et al. (2024). Closing the Loops on Southern Ocean Dynamics: From the Circumpolar Current to Ice Shelves and From Bottom Mixing to Surface Waves. *Reviews of Geophysics*, 62(3), Article e2022RG000781. <https://doi.org/10.1029/2022RG000781>
- Dutton, A., Carlson, A. E., Long, A. J., Milne, G. A., Clark, P. U., DeConto, R., Horton, B. P. et al. (2015). Sea-level rise due to polar ice-sheet mass loss during past warm periods. *Science*, 349(6244), aaa4019. <https://doi.org/10.1126/science.aaa4019>
- Elderfield, H., Ferretti, P., Greaves, M., Crowhurst, S., McCave, I. N., Hodell, D., & Piotrowski, A. M. (2012). Evolution of ocean temperature and ice volume through the mid-Pleistocene climate transition. *Science*, 337(6095), 704–709. <https://doi.org/10.1126/science.1221294>
- Hasenfratz, A. P., Jaccard, S. L., Martínez-García, A., Sigman, D. M., Hodell, D. A., Vance, D., Bernasconi, S. M. et al. (2019). The residence time of Southern Ocean surface waters and the 100,000-year ice age cycle. *Science*, 363(6431), 1080–1084. <https://doi.org/10.1126/science.aat7067>
- Hayes, C. T., Martínez-García, A., Hasenfratz, A. P., Jaccard, S. L., Hodell, D. A., Sigman, D. M., Haug, G. H. et al. (2014). A stagnation event in the deep South Atlantic during the last interglacial period. *Science*, 346(6216), 1514–1517. <https://doi.org/10.1126/science.1256620>
- Hellmer, H. H., Kauker, F., Timmermann, R., Determann, J., & Rae, J. (2012). Twenty-first-century warming of a large Antarctic ice-shelf cavity by a redirected coastal current. *Nature*, 485(7397), 225–228. <https://doi.org/10.1038/nature11064>
- International Organization for Standardization. (1993). *ISO Standards Handbook: Quantities and units* (3rd ed.). Geneva: ISO.
- Mas e Braga, M., Bernales, J., Prange, M., Stroeven, A. P., & Rogozhina, I. (2021). Sensitivity of the Antarctic ice sheets to the warming of marine isotope substage 11c. *The Cryosphere*, 15(1), 459–478. <https://doi.org/10.5194/tc-15-459-2021>
- Orsi, A. H., & Wiederwohl, C. L. (2009). A recount of Ross Sea waters. *Deep Sea Research Part II: Topical Studies in Oceanography*, 56(13-14), 778–795. <https://doi.org/10.1016/j.dsr2.2008.10.033>
- Prend, C. J., MacGilchrist, G. A., Manucharyan, G. E., Pang, R. Q., Moorman, R., Thompson, A. F., Griffies, S. M. et al. (2024). Ross Gyre variability modulates

- oceanic heat supply toward the West Antarctic continental shelf. *Communications Earth & Environment*, 5(1), 47. <https://doi.org/10.1038/s43247-024-01207-y>
- Rintoul, S. R., Hughes, C. W., & Olbers, D. (2001). The Antarctic Circumpolar Current System. In G. Siedler, J. Church, & J. Gould (Eds.), *Ocean Circulation and Climate: Observing and Modelling the Global Ocean* (Vol. 77, 271-302). Academic. [https://doi.org/10.1016/S0074-6142\(01\)80124-8](https://doi.org/10.1016/S0074-6142(01)80124-8)
- Schröder, M., & Fahrbach, E. (1999). On the structure and the transport of the eastern Weddell Gyre. *Deep Sea Research Part II: Topical Studies in Oceanography*, 46(1-2), 501–527. [https://doi.org/10.1016/S0967-0645\(98\)00112-X](https://doi.org/10.1016/S0967-0645(98)00112-X)
- Solodoch, A., Stewart, A. L., Hogg, A. M., Morrison, A. K., Kiss, A. E., Thompson, A. F., Purkey, S. G. et al. (2022). How does Antarctic bottom water cross the Southern Ocean? *Geophysical Research Letters*, 49(7), e2021GL097211. <https://doi.org/10.1029/2021GL097211>
- Thompson, A. F., Stewart, A. L., Spence, P., & Heywood, K. J. (2018). The Antarctic slope current in a changing climate. *Reviews of Geophysics*, 56(4), 741–770. <https://doi.org/10.1029/2018RG000624>
- Timmermann, A., Friedrich, T., Timm, O. E., Chikamoto, M. O., Abe-Ouchi, A., & Ganopolski, A. (2014). Modeling Obliquity and CO₂ Effects on Southern Hemisphere Climate during the Past 408 ka *Journal of Climate*, 27(5), 1863–1875. <https://doi.org/10.1175/JCLI-D-13-00311.1>
- Vimeux, F., Cuffey, K. M., & Jouzel, J. (2002). New insights into Southern Hemisphere temperature changes from Vostok ice cores using deuterium excess correction. *Earth and Planetary Science Letters*, 203(3-4), 829–843. [https://doi.org/10.1016/S0012-821X\(02\)00950-0](https://doi.org/10.1016/S0012-821X(02)00950-0)

Review: “Southern Ocean Evidence for Recurring West Antarctic Ice Sheet Destabilization During Marine Isotope Stage 11”

Authors: L. Jebasinski, D. A. Frick, A. K. I. U. Kapuge, C. Basak, M. Saavedra-Pellitero, G. Winckler, F. Lamy, J. Gottschalk

Summary

This study uses a multi-proxy approach to identify recurring millennial-scale deoxygenation episodes in AABW in the central Pacific during MIS 11, concurrent with similar events in the Atlantic. These synchronized AABW stagnation events are linked to WAIS retreat driven by subsurface warming from CDW. The results suggest a circumpolar response to ocean heat, with WAIS meltwater contributing significantly to MIS 11 sea-level high stands and potentially to future sea-level rise under continued warming.

The authors present interesting and timely results, highlighting the physical and dynamical roles of WAIS meltwater and CDW in driving recurring AABW deoxygenation events, with relevance to ongoing and future climate change. Overall, I find this to be a well-written and well-structured paper. However, I have several major concerns, particularly regarding the inconsistent proxy patterns across the proposed AABW stagnation events and the interpretation of key physical processes, as detailed below.

Major comments

- (1) **Recurring events:** I appreciate the authors' use of a multi-proxy approach in Figures 2 and 3, which reveals generally good agreement among several variables during MIS 11. However, not all proxies exhibit consistent recurring patterns across the EI–EIII events. For example, summer SST/water isotope (δD) and benthic foraminiferal $\delta^{18}O$ (Fig. 2), as well as BWT and relative sea level (Fig. 3), do not show clearly repeated signals. Could the authors clarify the reasons for these discrepancies? In addition, the term “AABW stagnation events” is not clearly defined in the manuscript. At times, two events are mentioned; at other times, three, depending on the interpretation of the IODP site data. I suggest the authors clearly specify how many such events are explicitly discussed and on what criteria

this identification is based.

- (2) **Negative feedbacks between WAIS retreat and AABW contraction:** I have a concern regarding the authors' statement that "*meltwater supply into the Southern Ocean might have steepened the isopycnal gradient along the shelf break*" (lines 376–377). Whether such steepening occurs likely depends on how effectively the meltwater is retained on the continental shelf. Given the large volume of meltwater input (~ 50 m sea-level equivalent, as shown in Fig. 3i), a substantial portion would likely have escaped to the open ocean. In that case, the West Antarctic shelf would not remain isolated, and the isopycnal gradient along the shelf break would more likely be flattened rather than steepened.
- (3) **Development of an open-ocean polynya:** I do not agree with the authors' statement that "*expanded Antarctic sea ice and northward shifted westerly winds could also have promoted the development of an open-ocean polynya favoring deep-reaching convection*" (lines 381–383). While I understand that this conclusion is based on the assumption of "*a northward shift of the westerlies*", it is important to note that this occurred under warming conditions (as indicated by SST in Fig. 2e). In general, open-ocean polynyas are expected to decrease in a warmer climate due to enhanced stratification and reduced sea ice formation.
- (4) **CDW upwelling in Ross Sea (Fig. 4c):** I cannot follow the authors' statement "*warm CDW intrusions onto the continental shelf of the Ross Sea Embayment, presumably along the eastern limb of the Ross Gyre*" (lines 399–401). Are the authors referring to CDW itself, or to the Antarctic Slope Current (ASC) as the transport mechanism? I do not understand how CDW would move along the Ross Gyre as shown in Fig. 4c.

Minor comments

- In Sections 2.1 and 2.2, please refer specifically to the subpanels in Figures 2 and 3 (e.g., Fig. 2a, 2b, Fig. 3a, 3b, etc.) where appropriate. Given the large number of subplots, it is difficult for readers to follow the discussion without clear and direct references to the relevant panels.
- Figure 1b: I am not sure it is appropriate to plot tracer concentrations originating from both

the Weddell Sea and the Ross Sea on the same figure. The overlapping region within the Weddell Sea makes the results difficult to interpret.

- Figure 2d caption: Change “*age model (see Results)*” to “age model (see Methods)”.